# Genome of the estuarine oyster provides insights into climate impact and adaptive plasticity

Ao Li[1,2,8], He Dai[3,8], Ximing Guo [4,8], Ziyan Zhang[1,5,6], Kexin Zhang[1,5,6], Chaogang Wang[1,5,6], Xinxing Wang[1,5,6], Wei Wang[1,5,7], Hongju Chen[3], Xumin Li [3], Hongkun Zheng [3], Li Li [1,5,6,7 ✉] & Guofan Zhang [1,2,7 ✉]

Understanding the roles of genetic divergence and phenotypic plasticity in adaptation is central to evolutionary biology and important for assessing adaptive potential of species under climate change. Analysis of a chromosome-level assembly and resequencing of individuals across wide latitude distribution in the estuarine oyster (*Crassostrea ariakensis*) revealed unexpectedly low genomic diversity and population structures shaped by historical glaciation, geological events and oceanographic forces. Strong selection signals were detected in genes responding to temperature and salinity stress, especially of the expanded *solute carrier* families, highlighting the importance of gene expansion in environmental adaptation. Genes exhibiting high plasticity showed strong selection in upstream regulatory regions that modulate transcription, indicating selection favoring plasticity. Our findings suggest that genomic variation and population structure in marine bivalves are heavily influenced by climate history and physical forces, and gene expansion and selection may enhance phenotypic plasticity that is critical for the adaptation to rapidly changing environments.

[1] CAS and Shandong Province Key Laboratory of Experimental Marine Biology, Center for Ocean Mega-Science, Institute of Oceanology, Chinese Academy of Sciences, Qingdao, China. [2] Laboratory for Marine Biology and Biotechnology, Pilot National Laboratory for Marine Science and Technology, Qingdao, China. [3] Biomarker Technologies Corporation, Beijing, China. [4] Haskin Shellfish Research Laboratory, Department of Marine and Coastal Sciences, Rutgers University, Port Norris, NJ, USA. [5] Laboratory for Marine Fisheries Science and Food Production Processes, Pilot National Laboratory for Marine Science and Technology, Qingdao, China. [6] University of Chinese Academy of Sciences, Beijing, China. [7] National and Local Joint Engineering Key Laboratory of Ecological Mariculture, Institute of Oceanology, Chinese Academy of Sciences, Qingdao, China. [8] These authors contributed equally: Ao Li, He Dai, Ximing Guo. ✉email: lili@qdio.ac.cn; gzhang@qdio.ac.cn

C limate change poses a serious threat to global biodiversity and the stability of ecosystems. Oceans bear the blunt of climate change as they absorb most of the heat energy from the sun and about one-third of the carbon dioxide created by human activities[1]. World oceans become warmer and more acidic. Sea levels are rising, and ocean circulation patterns are changing. With the rising of global temperature, dry regions are getting drier, while wet regions become wetter[2], increasing salinity difference among estuaries[3]. These rapid changes in world's oceans are having a great impact on marine ecosystems, much of which remains poorly understood. The impact may vary among different organisms, and changes in keystone species may have a large effect on stability of an ecosystem. Understanding how organisms adapt to environmental shifts is fundamental to evolutionary biology and critical in assessing their adaptive potential under climate change.

The ability of an organism to survive and adapt to sudden environmental changes depends on available genetic variation and phenotypic plasticity. The standing genetic variation of a species is the outcome of its evolutionary history shaped by mutation, selection and genetic drifts[4–6]. Past and on-going climate and geological events may leave signatures of selection or bottleneck in the genome and influence the geographic distribution of genetic variation[7–9]. Many recent studies have revealed the important role of phenotypic plasticity in environmental adaptation[10–15]. Phenotypic plasticity may buffer against sudden environmental changes and provides time for adaptation to occur[12–14,16]. Phenotypic plasticity is particularly pronounced and important for sessile organisms inhabiting estuarine and intertidal zones that cannot use avoidance to cope with wide fluctuations in environmental conditions[11,12,17], although the genetic basis of the enhanced plasticity is not clear[18,19].

Oysters are keystone species in coastal and estuarine ecology providing critical ecological services as filter-feeders and habitat engineers. As sessile bivalves thriving in the coastal zone, oysters are well adapted to highly dynamic environmental conditions, as reflected in their high genetic diversity and plasticity[12,20–23], making them interesting models for studying responses to climate change. In addition to threats from climate change, oyster populations worldwide have been decimated by over-fishing, habitat destruction and infectious diseases[24,25]. The estuarine (Jinjiang or Suminoe) oyster (*Crassostrea ariakensis*, Fig. 1a) is broadly distributed in estuaries of East Asia over large latitude and experiences wide ranges of temperature and salinity[24–28]. Unlike the Pacific oyster (*Crassostrea gigas*), a sympatric sister-species that is highly abundant in high-salinity waters of higher latitude, the estuarine oyster is only found in lower-salinity estuaries within its wide distribution range and at low abundance. The evolutionary pathways leading to the stark differences in abundance, distribution and salinity preference are not clear but important to our understanding of adaptive evolution.

While genetic changes can be inferred with genetic markers, whole-genome analysis is essential for exploring all genetic variation and identifying genes and selection events that are critical for adaption. Using multi-omic analyses, we previously showed that the Pacific oyster has a highly polymorphic genome with extensive expansion of environment-responsive genes, and plasticity is positively correlated with local adaptation[12,23]. In this study, we produced a chromosome-level assembly of the estuarine oyster genome, re-sequenced 264 wild oysters collected from 11 estuaries, and conducted transcriptomic studies of environmental response to understand its genetic variation, population structure, phenotypic plasticity and genomic signatures of selection or bottleneck that may be linked to environmental changes. Our results show that the estuarine oyster has significantly lower genetic diversity than its sister-species probably due to the impact of past

glaciation on its unique estuarine lifestyle. Its population structure is heavily influenced by geological events and ocean currents. Expansion and selection in regulatory regions of environment-responsive genes may enhance phenotypic plasticity that is critical for the adaptation to rapidly changing environments.

## Results and discussion

**Genome assembly and annotation.** We produced a chromosome-level assembly of the estuarine oyster genome of an estimated size of 614.05 Mb (19-mer analysis) (Supplementary Fig. 1, Supplementary Table 1), using a combination of nanopore long-reads generated from nine flow cells on the PromethION platform (184 Gb, 299.24×), Illumina paired-end short-reads (64 Gb) and Hi-C (106 Gb) sequences and a hierarchical assembly approach. The final assembly consisted of 630 contigs with a N50 of 6.97 Mb, spanning 613.89 Mb, 99.6% of which or 416 scaffolds with a scaffold N50 of 62.26 Mb were assembled into 10 chromosomes corresponding to the haploid number (Fig. 1b and c, Supplementary Table 2). To our knowledge, this is the most contiguous assembly (by contig N50) produced with nanopore long-read sequencing for a bivalve mollusc[29–38] (Supplementary Table 3). The coverage of the assembled genome was assessed by mapping RNA-seq reads and Illumina genomic reads, and over 97.9% of genomic short-reads and 97.2% RNA-seq reads were mapped to the assembly respectively (Supplementary Table 4). Also, the genome assembly captured 92.24% of the Benchmarking Universal Single Copy Orthologs (BUSCO) datasets (Fig. 1d, Supplementary Data 1), indicating the assembly has a high level of completeness. The accuracy of genome sequence was 98.32% as compared with Sanger sequencing (Supplementary Table 5). Analysis with REAPR showed that both fragment coverage distribution error and low fragment coverage over a gap were zero, indicating that the genome assembly was accurate. These results indicate that the estuarine oyster genome assembly is of high quality and can be used for down-stream analyses.

The estuarine oyster genome encoded 29,631 protein-coding genes as indicated by homology, *de novo* prediction and mRNA transcripts, 96.13% of which were functionally annotated (Supplementary Table 6). Various non-coding RNA sequences were also identified and annotated in the genome, including 1,077 transfer RNAs, 20 microRNAs and 131 ribosomal RNAs. A total of 332.40 Mb (54.14%) of repetitive elements were identified (Supplementary Table 7), which is higher than the 43% observed in the Pacific oyster[32]. Gene density was inversely correlated to the repetitive element content across all chromosomes (Supplementary Fig. 2, $\rho = -0.308$, $p < 0.001$). Genome-wide polymorphism was 0.58%, which is less than half as that of the Pacific oyster[23], probably due to population bottleneck caused by past glaciations more severely impacting its estuarine lifestyle (see below).

We analyzed macrosynteny between *C. ariakensis* and two other oyster species inhabiting low-salinity estuarine, *C. virginica* and *C. hongkongensis*. High colinearity was found between *C. ariakensis* and *C. hongkongensis* across 205.48 Mb covering 20,571 genes of *C. ariakensis* genome (Supplementary Data 2), while a lower colinearity between *C. ariakensis* and *C. virginica* was detected across 194.17 Mb covering 18,692 genes of *C. ariakensis* genome (Supplementary Fig. 3, Supplementary Data 3). Our findings are consistent with the fact that the two Asian species *C. ariakensis* and *C. hongkongensis* have a closer phylogenetic relationship that diverged 22.3 Myr ago, while the Atlantic *C. virginica* diverged from the Asian species about 82.7 Myr ago[39]. Assembly errors in the *C. virginica* genome as indicated by the discrepancy with linkage map (X. Guo, personal communication) may also explain some of the differences.

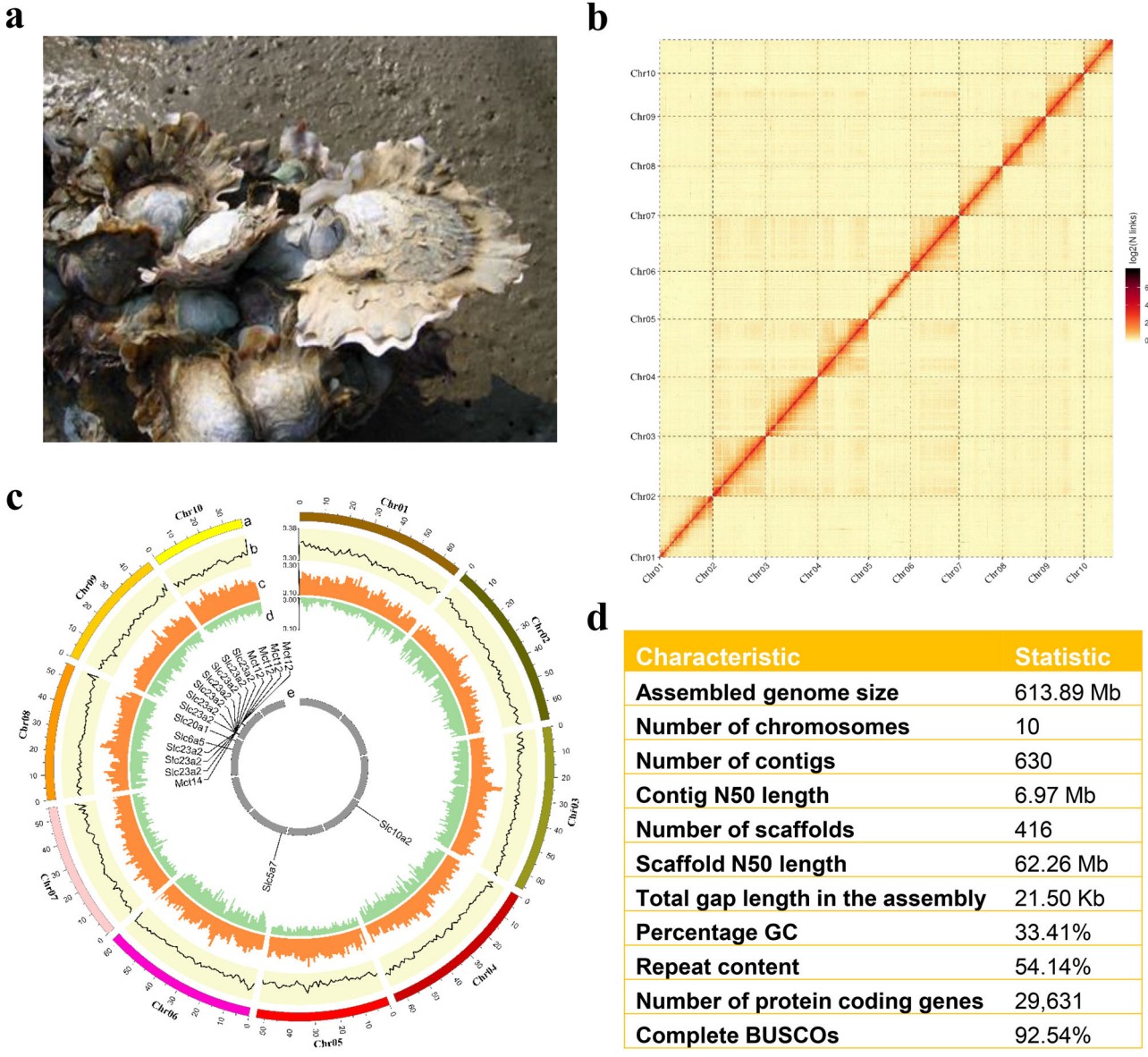

**Fig. 1 Chromosome-level assembly of the estuarine oyster *Crassostrea ariakensis* genome. a** Estuarine oyster (photo by Lumin Qian). **b** Hi-C interaction heatmap showing 10 chromosomes of the estuarine oyster. **c** CIRCOS plot showing 10 chromosomes (a), the distribution of GC content (b), transposable elements (c), coding sequences (d), and duplicated gene cluster of the *solute carrier* families showing selection signals (e, also see Supplementary Fig. 13). **d** Summary statistics of the genome assembly.

| Characteristic | Statistic |
| --- | --- |
| **Assembled genome size** | 613.89 Mb |
| **Number of chromosomes** | 10 |
| **Number of contigs** | 630 |
| **Contig N50 length** | 6.97 Mb |
| **Number of scaffolds** | 416 |
| **Scaffold N50 length** | 62.26 Mb |
| **Total gap length in the assembly** | 21.50 Kb |
| **Percentage GC** | 33.41% |
| **Repeat content** | 54.14% |
| **Number of protein coding genes** | 29,631 |
| **Complete BUSCOs** | 92.54% |

## Genome resequencing, variation calling and population structure

We generated 3.81 Tb clean whole-genome resequencing data from 264 wild oysters collected from 11 estuaries covering most of the distribution range[26,28] (Fig. 2a). The genome mapping rate averaged 95.3% ranging from 86.5% to 96.7%, and the mapped read depth averaged 19.89×. The data generated 145,271,754 SNPs (ranging from 487,881 to 640,962 per individual) and 103,080,822 indels (ranging from 342,486 to 443,381 per individual). Overall, there were 0.47 heterozygous SNPs per Kb per individual (Supplementary Data 4), which is ~35-fold lower than that in Pacific oyster populations[12].

Structure analysis of genome-wide SNPs revealed significant differentiation among different geographic populations, consistent with previously analyses of fitness-related traits, neutral markers and transcriptomic data[26,40–42], but with higher resolution. The optimal number of population clusters was identified as $k = 3$ (Supplementary Fig. 4), representing three main regions along the coast of northern China (NC, 5 sites), middle China (MC, 2 sites) and southern China (SC, 4 sites) (Supplementary Fig. 5). Principle component analysis (PCA), explaining 16% of genetic variance by two PCs, consistently revealed three distinct populations corresponding to NC, MC and SC. Further, fine-scale subpopulations were detected where oysters from Qingdao (QD) and southern subpopulation (SC-b) including Taishan (TS) and Qinzhou (QZh) were separated from NC and SC, respectively (Fig. 2b). Moreover, phylogenetic analysis using the Maximum Likelihood (ML) method supported PCA clustering, by first distinguishing southern populations from others and then identified a subpopulation (NC-a) within NC located in estuaries of southern Bohai Sea including Binzhou (BZ) and Dongying (DY) (Fig. 2c). A total of six subpopulations were identified for 11 estuaries, greatly improving the resolution of the fine-population structure. Pairwise $F_{ST}$ using all polymorphic positions revealed exceptionally high divergence between SC and other populations, ranged from 0.143 to 0.225 (Fig. 2d), while divergence within SC, MC and NC populations was much lower (Supplementary

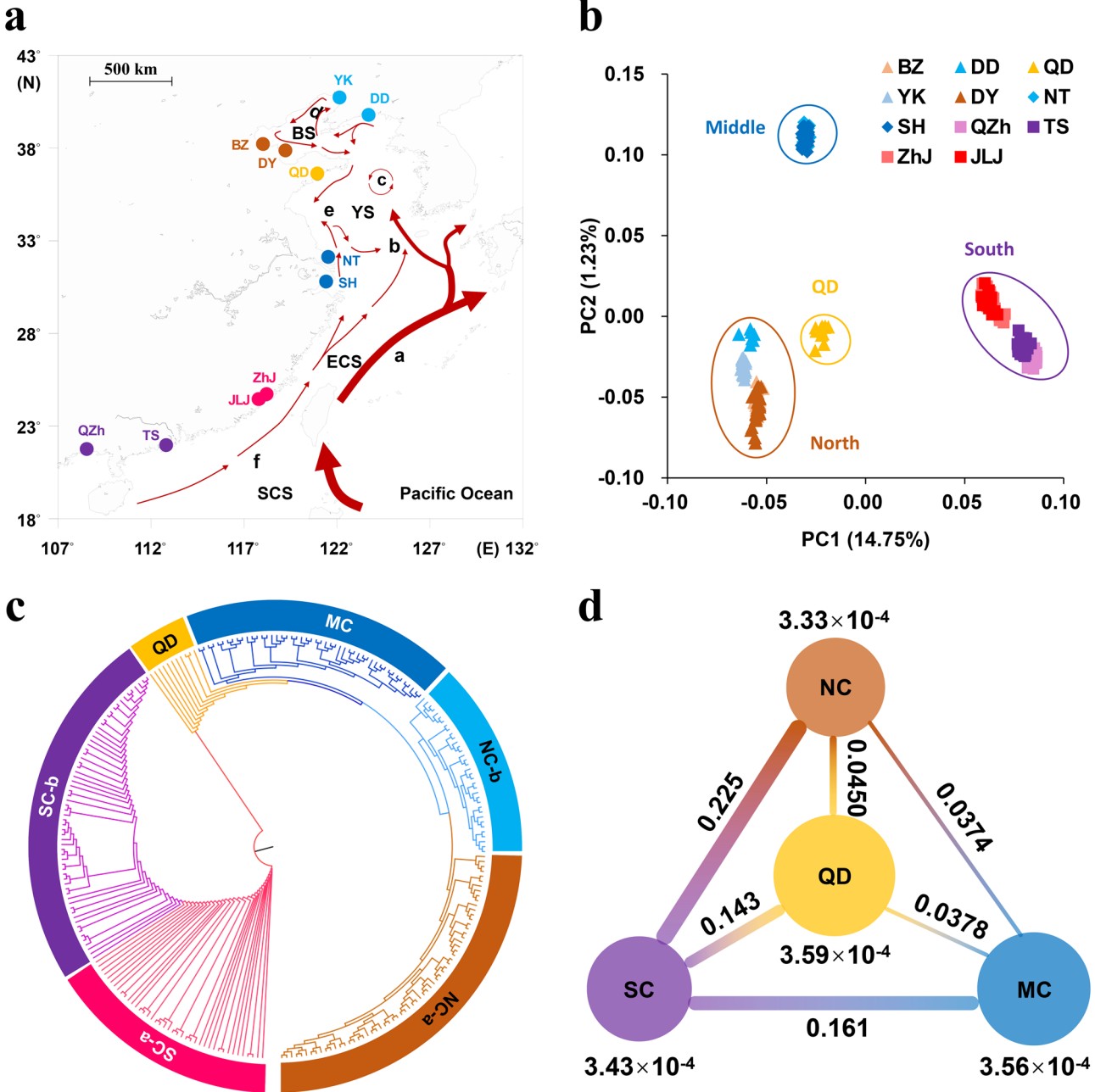

**Fig. 2 Geographic distribution and genetic differentiation of the estuarine oyster populations. a** Sampling locations of 264 resequenced wild oysters from 11 estuaries along the coast of China. The arrowed curves represent ocean currents in summer. a: Kuroshio Current, b: Yellow Sea Warm Current, c: Yellow Sea Cold Water Mass, d: Bohai Sea Circulation, e: China Coastal Current, f: South China Sea Warm Current. SCS: South China Sea, ECS: East China Sea, YS: Yellow Sea, BS: Bohai Sea. DD: Dandong, YK: Yingkou, BZ: Binzhou, DY: Dongying, QD: Qingdao, NT: Nantong, SH: Shanghai, JLJ: Jiulongjiang, ZhJ: Zhangjiang, TS: Taishan, QZh: Qinzhou. **b** Plots of principal components 1 and 2 of 264 resequenced oysters based on whole-genome data. **c** Phylogenetic tree of estuarine oysters inferred from genome-wide SNPs with the Maximum Likelihood (ML) method. NC-a: northern China, including BZ and DY; NC-b: northern China including DD and YK; MC: middle China including NT and SH; SC-a: southern China including JLJ and ZhJ; and SC-b: southern China including TS and QZh. **d** Nucleotide diversity (*pi*, under or above the circles) and genetic divergence ($F_{ST}$, between populations) among four populations.

Table 8). Oysters from MC and NC were clustered together in phylogenetic tree with a lower genetic divergence ($F_{ST} < 0.05$), which is comparable with the divergence among populations of the Pacific oyster in north China[12]. Linkage disequilibrium (LD, measured as $r^2$) decreased to half of its maximum value around $2.54 − 3.00$ kb in the three populations (Supplementary Fig. 6), which is substantially slower than the LD decay in the Pacific oyster (~0.1 kb)[12]. These results provide unprecedented insights into the fine-scale population structure of the estuarine oyster.

The finding of exceptionally low sequence diversity, slow LD decay and high population divergence in the estuarine oyster compared with the Pacific oyster suggests that the two species experienced different evolutionary forces.

**Historical glaciation, geological events and oceanographic forces shaping diversity and distribution.** Population structure of the estuarine oyster was largely concordant with the direction

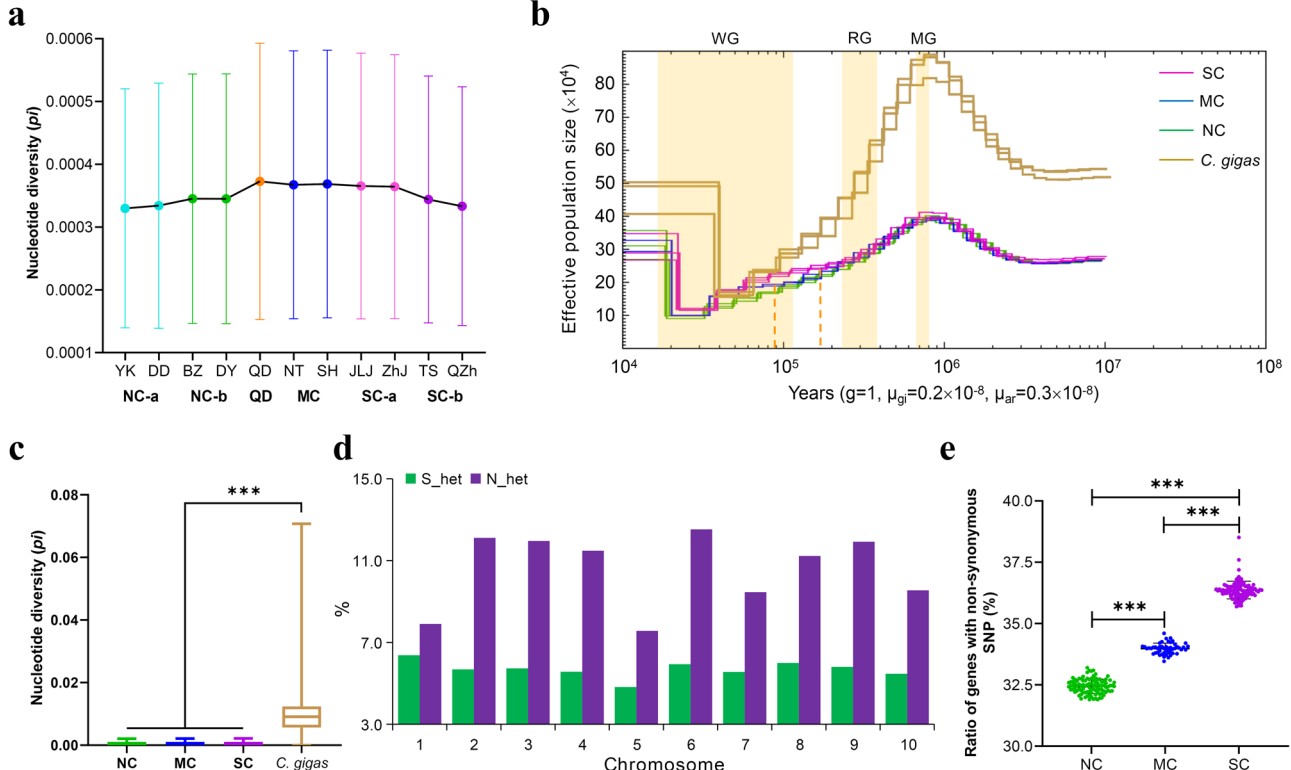

**Fig. 3 Effects of gene flow, historical glaciation and natural selection on diversity and differentiation of the estuarine oyster.** **a** Nucleotide diversity (*pi*) in 11 geographic populations along the coast of China. **b** Demographic histories of three populations of the estuarine oyster (ar) (SC, MC and NC), and Pacific oyster (*C. gigas* or gi) inferred by Pairwise Sequentially Markovian Coalescent. Periods of the Mindel glaciation (MG, 0.68~0.80 mya), Riss glaciation (MG, 0.24~0.37 mya) and Würm glaciation (WG, 10,000~120,000 years ago) were shaded by yellow. **c** Nucleotide diversity (*pi*) of three populations of the estuarine oyster, and Pacific oyster (*C. gigas*). The data are presented as box plots; the central rectangle spans the first to third quartiles of the distribution, and the 'whiskers' above and below the box show the maximum and minimum estimates. The line inside the rectangle shows the median, the circles represent outliers. **d** The ratio of SNPs that are highly heterozygous only in northern (N_het) or only in southern (S_het) populations across 10 chromosomes. **e** The ratio of genes with non-synonymous SNPs in three oyster populations. Data are presented as circle plot, with each circle representing an individual oyster and bars showing the mean value ± SEM. Asterisks indicate significant difference (\*\*\**p* < 0.001). Error bars represent SEM values.

of ocean currents during the summer, where the northern or southern coastal currents are not crossed except over middle populations near the Yangtze River estuary (Fig. 2a and Supplementary Movie 1). Nucleotide diversity (*pi*) was highest in middle populations ($3.56 \times 10^{-4}$) and QD oysters ($3.59 \times 10^{-4}$), likely because converging southern and northern currents bring increased genetic diversity to the middle populations (Fig. 3a). These findings indicate that ocean currents play important role in shaping and maintaining population structure of the estuarine oyster[12,43]. The exceptionally high divergence between SC and NC populations can be explained by the fact that currents from the south and north do not cross over and cannot facilitate gene flow. Freshwater discharge from Yangtze River is considered as a major barrier limiting gene flow between north and south populations of marine organisms[7,44–47]. For the estuarine oyster, however, populations on the north (NT) and south (SH) side of Yangtze River were clustered together as MC, indicating Yangtze River is not a barrier to gene flow for the estuarine oyster that thrives in low salinity water of estuaries.

To reconstruct the demographic history, Pairwise Sequentially Markovian Coalescent (PSMC) method were used to assess fluctuations in effective population size (*Ne*) in response to Quaternary climatic change using genome data of two or three estuarine oysters from each of the three populations, as well as three previously sequenced Pacific oysters for comparisons[12]. Both species were severely affected by glaciation events during the

past million year as their *Ne* peaked at ~0.90 mya before the Mindel glaciation (MG, 0.68~0.80 mya) and then substantially decreased over three periods of glaciation: the MG, Riss (RG, 0.24~0.37 mya) and Würm (WG, 10,000~120,000 years ago) glaciation. The *Ne* of both species reached their bottom during the last glaciation WG, but Pacific oyster populations rebounded early and to higher levels than the estuarine oyster, suggesting the last ice age had a greater impact on the estuarine oyster. Also, during the interglacial period before the last glaciation at ~0.2 mya, the SC and NC populations of the estuarine oyster started to diverge (Fig. 3b). We also estimated divergence time between SC and NC based on cytochrome oxidase I (COI) sequence data using divergence time between *C. gigas* and *C. angulata* as reference[39,46]. The COI-based estimate placed the divergence time at 0.14~0.63 mya, which is close to the 0.2 mya estimated by PSMC with whole-genome data (Fig. 3b). The divergence time coincided with the formation land bridge between Taiwan and mainland China from 0.2 mya to 25,000 years ago by tectonic movement[48], creating a physical barrier to gene flow between southern and northern populations that is now maintained by ocean currents.

The nucleotide diversity of all three populations of the estuarine oyster was 25-fold lower than that of the Pacific oyster ($pi_{\_C. \, gigas} = 9.27 \times 10^{-3}$, Fig. 3c). Similarly, stickleback fishes adapted to freshwater areas exhibited lower *pi* and *Ne* values than counterparts dwelling in brackish areas[49]. These findings suggest

that climate change may affect species with different lifestyles differently and the estuarine oyster may be more sensitive to climate change than the Pacific oyster that inhabits more open waters of high salinity. Further, $Ne$ curves of middle and northern populations split ~90,000 years ago, which corresponded to the great sea level fall at the sub-glaciation of WG when Bohai Sea dried up[50]. The lower $Ne$ and nucleotide diversity of northern population (Fig. 3a, b) support that a stronger bottleneck occurred in estuarine oysters of Bohai Sea. Relatively lower nucleotide diversity was also found in Bohai populations of the Pacific oyster[12], possibly due to the same sea level fall. The role of bottlenecks and geographic isolation resulting from historical glaciations and tectonic events in shaping phylogeography has been demonstrated in other molluscs, such as the divergence between Atlantic coast and Gulf of Mexico populations of the eastern oyster *Crassostrea virginica*[9]. Our results provide putative divergence times among estuarine oyster populations linked to historical events and suggest past climate and tectonic movement played an important role in shaping genetic diversity and population differentiation in the estuarine oyster.

To explore the effects of selection and genetic drift, we compared highly heterozygous SNPs (heterozygosity > 0.5) in northern and southern populations. The number of SNPs that are highly heterozygous in NC only (14,373) were 1.89-fold higher than those highly heterozygous in SC only (7,595) across 10 chromosomes (Fig. 3d). Both directional selection and genetic bottleneck can decrease the number of heterozygous SNPs. The finding that southern oysters had significantly higher ratio of genes with non-synonymous SNPs (35.67% ± 0.35%) than the middle (33.33% ± 0.22%) and northern (31.82% ± 0.29%) oysters ($p < 0.01$, Fig. 3e), suggests that the southern oysters experienced stronger selection. Stronger selection, potentially from strong environmental disturbance, decreased the numbers of heterozygous SNPs in southern oysters. Thus, in addition to isolation and bottlenecks from historical glaciation, geological events and oceanographic forces, selection or local adaptation may also play an important role in shaping variation and phylogeography of the estuarine oyster.

**Signatures of selection**. Oysters inhabiting northern and southern estuaries experience significantly different environmental conditions. Southern habitats are characterized by high temperature and low salinity. Satellite remote sensing data from 2000 to 2017 indicated that the monthly average sea surface temperature (SST) of southern habitats was 10.35 °C higher than that of northern habitats (Supplementary Fig. 7). Salinity at the northern BZ site was 10.98 ‰ higher than the southern TS site[3]. Climate change may enhance the difference and cause salinity to increase at north habitats but decrease at south habitats[51,52]. Thus, temperature and salinity are the two most important environmental factors driving adaptive divergence between northern and southern populations. With discontinuous distribution and limited gene flow, the southern population became locally adapted and evolved higher tolerance to high temperature and low salinity[3]. We expect that some genomic regions were subjected to selection and contributed to adaptation to higher temperature and lower salinity conditions in the southern population.

To identify signatures of selection sweeps, we calculated the fixation index ($F_{ST}$) and selection statistics (Tajima's D) between two population pairs (SC vs. NC and SC vs. MC) and identified $F_{ST}$ outliers (top 1%, $F_{ST}$ _north vs. south >= 0.693, $F_{ST}$_middle vs. south >= 0.637). Only genomic regions surrounding selective peaks that overlapped between the two population pairs and located at the valleys of Tajima's D values in one of the three

populations were considered as under selection. A total of 24 selective regions spanning 51 candidate genes (44 annotated) were identified along chromosomes 2, 3, 4, 6, 8 and 9 among three oyster populations (Fig. 4a, b and Supplementary Figs. 8–13, Supplementary Data 5). Most of these candidate genes are involved in response to environmental disturbances in salinity and temperature[53–60].

To assess transcriptomic response of the estuarine oyster to temperature and salinity disturbances, we conducted RNA-seq analysis of oysters exposed to elevated temperature (6 h under 37 °C) and high salinity (7 days under 60 ‰). Only genes with 10 or more aligned reads in >90% samples were used for subsequent analyses. About 44.17% (10,279 of 23,270) and 11.7% (2,757 of 23,650) of genes were differentially expressed under high salinity and temperature stresses, respectively, with 1,088 genes responsive to both stressors (Supplementary Fig. 14). For the 51 candidate genes from regions under selection, a total of 29 genes were expressed, 75.9% (22 of 29) and 58.6% (17 of 29) of which were differentially expressed under high salinity and high temperature, respectively ($p < 0.05$, Fig. 4c), indicating most genes under selection are involved in response to environmental disturbances in these two factors. Thirteen genes responded to both thermal and salinity stresses, while only three genes were not sensitive to the two stressors. Nine and four genes were highly responsive to high salinity and high temperature, respectively. The finding that most of genes from regions under selection are involved in response to temperature and salinity disturbances suggests that these two environmental factors are important drivers of adaptive evolution in the estuarine oyster.

**Expansion of genes contributing to temperature and salinity adaptation**. Among the 24 regions under selection, we found two clusters of tandemly duplicated genes belonging to *solute carrier* families, 10 copies of *Slc23a2* and four copies of *monocarboxylate transporter 12* (*Mct12*, also known as *Slc16a12*) (Supplementary Fig. 13c), located in two regions of chromosome 9 with high divergence ($F_{ST} = 0.81$ and 0.76) among the three populations (Supplementary Fig. 13d). The genomic region spanning *Slc23a2* gene families exhibited extremely low Tajima's D values in northern oysters, while these spanning *Mct12* gene families had extremely low Tajima's D values in southern oysters (Supplementary Fig. 13e). These findings indicate that *Slc23a2* and *Mct12* genes were under directional selection in northern and southern environments respectively, highlighting the critical role of *Slc* family genes in adaptation of marine species, as also reported in porpoises and coral[60–63], to salinity and temperature variations.

The ten copies of *Slc23a2* family belong to three orthogroups, where two of them are annotated as purine permease (a: OG0011985 and c: OG0000489) and the other is annotated as uric acid transporter (OG0000633). Four copies of *Mct12* gene family belong to one orthogroup annotated as purine efflux pump (OG0000571). All these orthogroups were extensively expanded in the estuarine oyster, as well as in the other two oyster species (*Crassostrea virginica* and *Crassostrea hongkongensis*) that occur in warm and low-salinity estuarine environments, but not so in *C. gigas* that inhabit cool, open and high-salinity waters (Supplementary Fig. 15). In addition to their presence in tandem arrays on chromosome 9, the expansion of the *solute carrier* families was further supported by phylogenetic analysis where duplicated genes were mostly clustered in ortholgroup- and lineage/species-specific manners (Supplementary Fig. 16). Gene copies that did not responds to heat and high-salinity stresses were not expressed or expressed at very lower expression (FPKM < 0.5) (Supplementary Table 9). Conversely, three copies of *Slc23a2* (*Slc23a2_a1*, *Slc23a2_a2* and *Slc23a2_c2*) in two expanded orthogroups

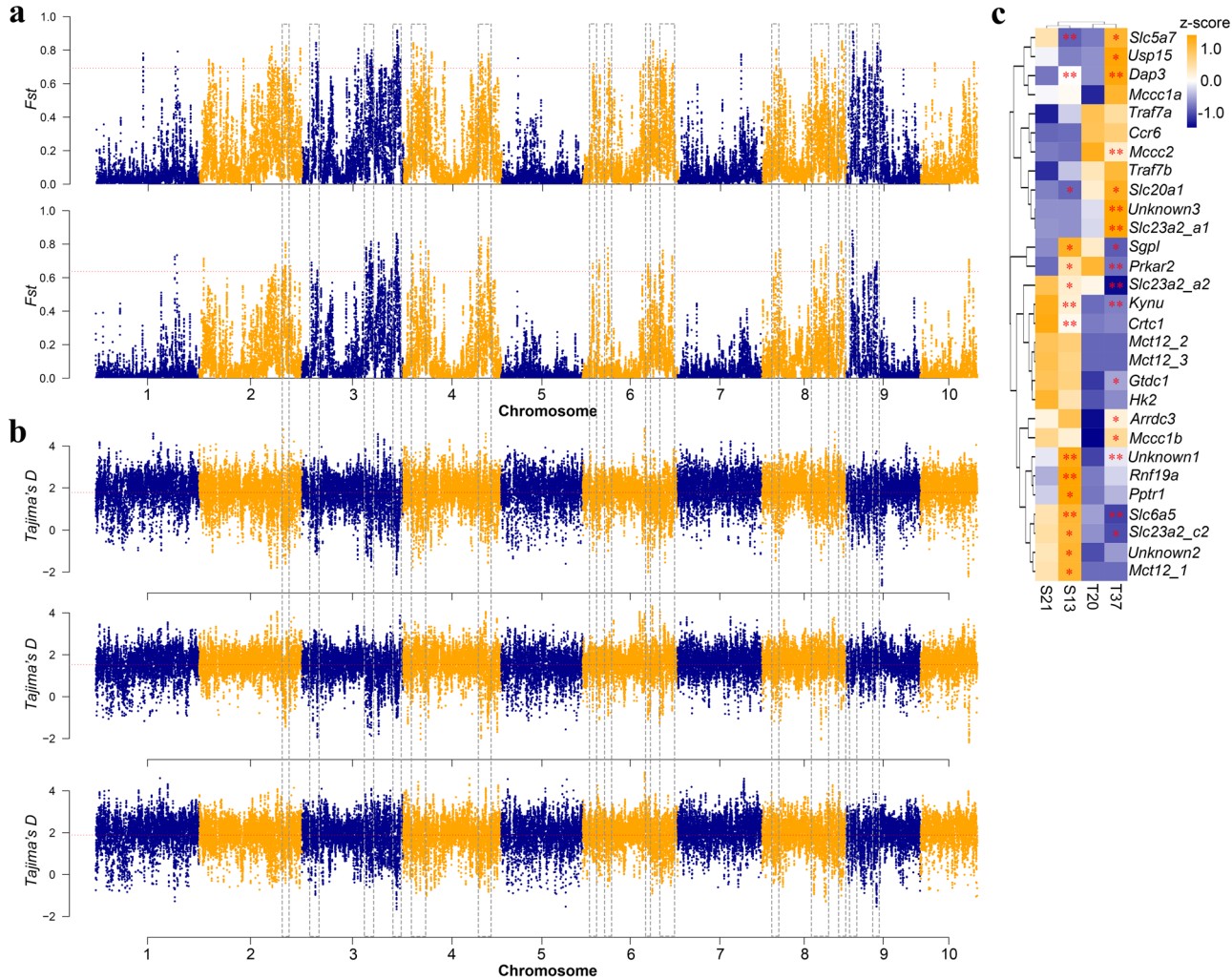

**Fig. 4 Genome-wide distribution of selective sweep signals and transcriptional responses of associated genes to thermal and salinity challenges in the estuarine oyster. a** Global $F_{ST}$ values (top 1%, red lines) between two population pairs: northern vs. southern (up) and middle vs. southern (bottom). **b** Global Tajima's $D$ values in northern (up), middle (middle) and southern (bottom) populations. **c** Expression of genes under selection exposed to thermal (6 h at 37 °C) and high-salinity (7 days at 60‰) challenges. Asterisks indicate significant difference (*$p < 0.05$, **$p < 0.01$). The blue-to-yellow color bar represents the relative expression level of the given gene from low to high.

responded to both temperature and salinity challenges, while three copies of *Mct12* (*Mct12_1*, *Mct12_2* and *Mct12_3*) responded to salinity challenge only (Fig. 4c) and preferentially clustered together (*Mct12_1/2/3* and *Slc23a2_a1/2*, Supplementary Fig. 16). The expansion of these environment-responsive genes may enhance transcriptional complexity or plasticity and play an important role in adaption to diverse temperature and salinity conditions in the estuarine oyster and two other species inhabiting similar environments. This finding provides further evidence that gene duplication is critical for stress adaptation, as previously demonstrated for the expansion of *heat shock protein* and *inhibitor of apoptosis* gene families in the Pacific oyster and other marine bivalves[23,35,64,65].

**Stronger selection in upstream regulatory regions of environment-responsive genes with high plasticity.** To understand the interaction between divergence and plasticity, we examined divergence in different genomic regions (genic, upstream and downstream) with $F_{ST}$ and transcriptional plasticity of the 29 genes from regions under selection in response to environmental translocations. $F_1$ progeny produced from northern and southern populations were acclimatized at both northern

and southern habitats for three months before transcriptional analysis. Fourteen of the 29 genes exhibited higher plasticity (HP) where they showed significantly differentiated expression when translocated to nonnative environments in both oyster populations ($p < 0.05$), while other genes had lower plasticity (LP) and their expression didn't strongly respond to environmental translocation/change (Fig. 5a). The transcriptional plasticity of the HP genes was 19.15% higher than that of LP genes ($p = 0.0255$, Supplementary Fig. 17).

For the whole genome, genic regions showed significantly higher $F_{ST}$ values (mean $F_{ST\_genic} = 0.163$) than that of intergenic regions ($F_{ST\_intergenic} = 0.138$, $p < 0.001$, Wilcoxon signed-rank tests, Fig. 5b), indicating stronger selection in genic regions driving differentiation between northern and southern populations. For the 29 genes from regions under selection, both genic ($F_{ST} = 0.7745$) and intergenic ($F_{ST} = 0.7585$) regions showed strong selection signals or divergence compared with all genes in the genome, and the difference between genic and intergenic regions was not significant ($p > 0.05$). For both genic and intergenic regions, genes with high plasticity showed significantly ($p = 0.0230$) higher divergence ($F_{ST} = 0.7816$) between northern and southern oysters than LP genes ($F_{ST} = 0.7274$) (Fig. 5b).

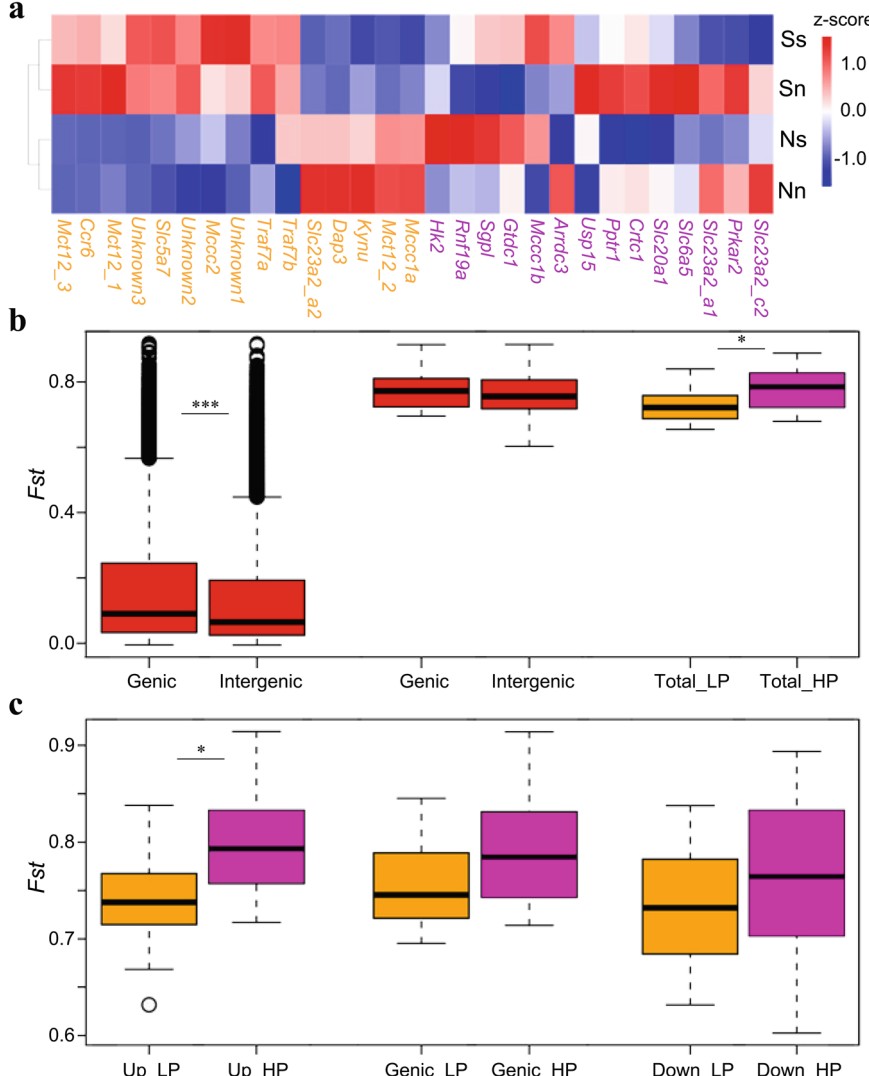

**Fig. 5 Transcriptional and genomic divergence of genes with selection signals in the estuarine oyster. a** Expression level of genes under selection in northern (N) and southern (S) populations (capital letters) acclimated to northern (n) and southern (s) environments (lowercase letters), showing low plasticity (LP, orange) and high plasticity (HP, purple) genes. The blue-to-red color bar represents the relative expression level of the given gene from low to high. **b** Genetic divergence ($F_{ST}$) for genic and intergenic regions at genome level (left) and for 29 genes under selection (middle), and for LP and HP genes in both genic and intergenic regions (Total, right). **c** Genetic divergence ($F_{ST}$) for intergenic [Up: upstream (left) and Down: downstream (right)] and genic (middle) regions of LP and HP genes. Asterisks indicate significant difference (*$p < 0.05$, ***$p < 0.001$). The data are presented as box plots; the central rectangle spans the first to third quartiles of the distribution, and the 'whiskers' above and below the box show the maximum and minimum estimates. The line inside the rectangle shows the median, the circles represent outliers.

Specifically, the upstream intergenic regions of HP genes exhibited significantly ($p = 0.01512$) higher divergence ($F_{ST} = 0.7958$) between northern and southern populations than that of LP genes ($F_{ST} = 0.7402$), while there was no significant difference between the two types of genes for both genic and downstream intergenic regions ($p > 0.05$, Fig. 5c). Further, LP genes showed higher divergence in genic regions than up and downstream regions. While selection preferentially targeted genic regions at genome level and for LP genes, HP genes showing high plasticity exhibited stronger selection in upstream intergenic regions, where critical regulatory elements such as promoter and enhancers reside, potentially regulate gene expression and enhance transcriptional plasticity[18]. These results suggest that selection and divergence in upstream regulatory regions may enhance phenotypic plasticity that contributes to the survival and adaption of organisms facing environmental disturbances[66]. The mechanism of selection favoring plasticity is not clear at this time. Directional

selection tends to reduce genetic variation and facilitate local adaptation. Selection acting on regulatory elements may create divergence and transcriptional complexity that enhance phenotypic plasticity[18,19] at species level. Directional selection acting on duplicated genes can potentially direct duplicated member genes to express under different conditions and therefore enhance overall phenotypic plasticity. It is also possible that balancing selection retains diversity in regulatory elements and increase plasticity. It has been suggested that balancing selection is pervasive in oysters and important for their evolution[67].

In conclusion, analysis of the estuarine oyster genome revealed exceptionally low genetic diversity and fine-scale population structures, which were shaped and maintained by climate history, geological events and oceanographic forces. Integration of genomic and transcriptomic analyses revealed genes in regions under selection are mostly involved in environmental response including the expanded *solute carrier* gene families that are

important for environmental adaptation. Expansion and selection in upstream regulatory regions of environment-responsive genes may hence be critical for the adaptation to rapidly changing environments in the estuarine oyster.

## Methods

**Genome sequencing.** A wild adult estuarine oyster *Crassostrea ariakensis* was obtained from Binzhou, Bohai Sea, northern China. Four tissue samples (gill, mantle, adductor muscle and labial palp) were collected and flash-frozen in liquid nitrogen. Genomic DNA was extracted from the adductor muscle using the DNeasy Blood & Tissue Kit (Qiagen, Hilden, Germany) and used to construct libraries for Oxford Nanopore Technologies' (ONT) long-read sequencing. gDNA (2 µg) was repaired using NEB Next FFPE DNA Repair Mix kit (M6630, USA) and subsequently processed using the ONT Template prep kit (SQK-LSK109, UK) according to the manufacturer's instructions. The large segment library was pre-mixed with loading beads and then pipetted into a previously used and washed R9 flow cell. The library was sequenced on the ONT PromethION platform with nine R9 cells and ONT sequencing reagent kit (EXP-FLP001.PRO.6, UK) according to the manufacturer's instructions. In addition, libraries with insert size of ~350 bp were prepared and sequenced using Illumina HiSeq 4000 platform.

**Genome size and heterozygosity.** We conducted k-mer copy number (KCN) analyses using k = 17, 19 and 21 to estimate genome size. K was the least odd number that conform to the following formula: $4^k/G > 200$, where G indicates the genome size (bp). Previous study estimated the genome size of *C. gigas* and *C. ariakensis* by flow cytometry, where their respective C-values, 0.89 pg and 0.99 pg[68], have a ratio of 1.112. Assuming the genome size of *C. gigas* is 586.8 Mb[34], the genome size of *C. ariakensis* would be 652.7 Mb. Thus, we selected 19-mer for genome size estimation.

Trimmed Illumina short reads were used to generate the KCN distribution. The 19-mer KCN distribution showed two distinct peaks (Supplementary Fig. 1). The first peak (KCN = 44) represents heterozygous single copy k-mers while the second peak (KCN = 89) represents homozygous single copy k-mers in the genome. Genome size was estimated by the formula G = K_num/peak depth.

**Contig-level assembly using long-reads.** Nanopore long reads, with a read N50 of 33,230 and a mean read length of 23,240 bp, were used for initial genome assembly. Error correction of clean data was conducted using Canu[69] v1.5, and then were assembled using Canu, WTDBG2[70] and SMARTdenovo tools. Quickmerge[71] v0.2.2 was used to join the three assemblies, and the assembly was corrected for 3 cycles with long reads using Racon[72] and for 3 cycles with Illumina reads using Pilon[73] v1.22 with default parameters (Supplementary Tables 10 and 11). Then, the Purge Haplotigs[74] software was used to remove redundant heterozygous contigs with parameter 'purge_haplotigs purge -a 55'.

**Chromosome-level assembly with Hi-C.** The same adductor muscle was used for crosslink with formaldehyde at room temperature and DNA was digested with HindIII. The 5' overhangs were adapted with biotinylated nucleotides, and free blunt ends were ligated. Then, the crosslinks were reversed and the DNA was purified and further sheared to 300-700 bp fragments. Streptavidin beads were used to isolate biotin-tagged fragments for PCR enrichment. The Hi-C library was constructed using Illumina's paired-end kits according to the manufacturer's instructions, which was subsequently sequenced on the Illumina HiSeq 4000 platform. Contigs and scaffolds which were then sorted and oriented into super scaffolds using LACHESIS[75] with the following parameters: cluster_min_re_sites = 47, cluster_max_link_density = 2, cluster_noninformative_ratio = 2, order_-min_n_res_in_trun = 40, order_min_n_res_in_shreds = 41.

**Genome evaluation.** Hi-C contact heatmap of the number of Hi-C links between 100 kb windows on the pseudochromosomes was used to assess the accuracy of the Hi-C assembly. Benchmarking Universal Single-Copy Orthologs (BUSCO, obd10) with 954 conserved genes were used to assess the completeness and accuracy of gene coverage. The completeness and accuracy of the assembly was also evaluated by mapping rate of RNA-seq reads (see Protein coding gene annotation) and Illumina genomic reads with Burrows-Wheeler Aligner[76] v.0.7.8. Further, the accuracy of genome sequences was assessed by REAPR[77] v. 1.0.18 with default parameters to identify errors in genome assemblies.

**Repeat annotation.** Transposable elements (TEs) were identified and classified using homology and *de novo*-based approaches. RepeatScout and LTR_FINDER were used to construct *de novo* repeat libraries. The *de novo*-based libraries were further classified with PASTEClassifier[78] to obtain a consensus library, and combined with the repeat library of Repbase data. RepeatMasker[79] v4.0.5 was used to identify TEs in the estuarine oyster genome with the combined library.

**Protein-coding gene annotation.** Protein-coding genes were identified with three methods: *de novo* prediction, homology-based prediction and mRNA-based

prediction. For *de novo* prediction, five ab initio gene prediction programs, Genscan[80] v1.0, Augustus[81] v2.4 (with transcriptomic data of the estuarine oyster used for training), GlimmerHMM[82] v3.0.4, GeneID[83] v1.4 and SNAP[84], were used to predict genes in the repeat-masked genome (hard-masking). For homolog-based prediction, protein sequences from 10 well-annotated species that were annotated by the NCBI Eukaryotic Genome Annotation Pipeline, *Homo sapiens*, *Danio rerio*, *Aplysia californica*, *Strongylocentrotus purpuratus*, *C. gigas*, *C. virginica*, *Biomphalaria glabrata*, *Lingula anatina*, *Octopus bimaculoides* and *Mizuhopecten yessoensis*, were downloaded from NCBI and aligned to the repeat-masked estuarine oyster genome using tblastn[85] with E-value ≤ 1E-05. We used GeMoMa[86] v1.3.1 to predict gene models based on alignment sequences. For mRNA-based prediction, four tissues (gill, mantle, adductor muscle and labial palp) were collected from the same oyster used for genome sequencing. Total RNA was extracted separately using TRIzol reagent (OMEGA, USA) according to the manufacturer's instructions. RNA quality and quantity were assessed using the Agilent 2100 instrument and Qubit 2.0 fluorometer (Thermo Fisher, USA), respectively. RNA from four tissues were combined in equimolar quantities into a single pool. Full-length coding DNA (cDNA) was obtained using SMARTer PCR cDNA Synthesis Kit (Clontech, USA). The library was constructed using SMRTbell® Template Prep Kit 2.0 (PacBio, USA). cDNA was sequenced by PacBio Sequel II platform with SMRT cell 8 M (Biomarker, China). RNA-seq data were filtered to remove adapters and then trimmed to remove low-quality bases. Clean reads were aligned to reference genome using TopHat2[87] and then assembled using Trinity[88]. Full transcriptome-based genome annotation was predicted using PASA[89] v2.2.2. Finally, EVidenceModeler[90] (EVM) v1.1.1 was used to generate a weighted and non-redundant gene set by integrating all gene models predicted by the three methods (Supplementary Table 12).

Homologous sequences in the genome were identified with genBlastA[91] v1.0.4 using the integrated gene set, and GeneWise[92] was used to identify pseudogenes. Transfer RNAs (tRNAs) were defined using tRNAscan-SE[93] v1.3.1 software with eukaryote default parameters. MicroRNA and rRNA were identified with Infernal BLASTN[95] against the Rfam[95] database v12.0.

Functional annotation of protein-coding genes was conducted by aligning them to the NCBI non-redundant protein[96] (NR), SwissProt[97], KOG[98] and TrEMBL[97] databases using BLAST[94] v2.2.31 with a maximal e-value of 1e-05. Domains were identified by searching against Pfam[99] database using HMMER[100] v3.0. Genes were mapped to Gene Ontology (GO) terms and KEGG pathways to identify their best functional classification.

**Synteny analysis.** We analyzed macrosynteny between *C. ariakensis* and the other two oyster species living in low-salinity estuaries, *C. virginica* and *C. hongkongensis*. First, we aligned gene sequences between two oyster species using Diamond[101] v0.9.29.130, to identify homologous gene pairs (e-value < 1e⁻⁵, C score > 0.5). MCScanX[102] was used to analyze chromosome collinearity for syntenic blocks between *C. ariakensis* and the other two oyster species (alignment significance = 1e⁻¹⁰), using Diamond result file and gff file. The maximum gap size was set to 25 genes and a minimum syntenic block required 5 genes.

**Whole-genome resequencing and mapping.** We collected 264 wild estuarine oysters from 11 estuaries (Fig. 3a), covering most of its distribution range[26,28]. Genomic DNA was isolated from gill tissue with standard phenol-chloroform extraction and used to construct paired-end libraries with an insert size of ~350 bp according to the manufacturer's instructions (Illumina Inc., San Diego, CA, USA) for sequencing on the Illumina HiSeq X Ten Sequencer. We obtained ~14.42 Gb of clean data for each sample, giving an average depth of 19.9X coverage (15-28X) (Supplementary Data 4 and 6). The 150-bp paired-end reads were mapped to the estuarine oyster reference genome (PRJNA715058) using BWA[76] with default parameters (bwa mem –M -t 10 -T 20). Mapping data were then converted into the BAM format and sorted by SAMtools v.1.3.1[103] to remove duplicate reads. Read pairs with the highest mapping quality were retained if multiple pairs have identical external coordinates.

**Variant calling and annotation.** The Genome Analysis Toolkit (GATK) v.3.7[104] module HaplotypeCaller was used to obtain high-quality variation calling of each sample. SNPs were further filtered with parameters 'QD < 2.0 || FS > 60.0 || MQ < 40.0'. Similarly, indels were called and filtered using parameters 'QD < 2.0 || FS > 60.0'. Filtered SNPs were annotated by SnpEff[105] and then classified into regions of exon, intron, splicing site, and upstream and downstream intergenic regions, and as heterozygous or homozygous variants. To characterize the types of variants in northern and southern oysters, Plink[106] was used to filter raw SNPs using parameters of MAF > 0.05 and Int > 0.8. The retained SNPs were singly classified as highly heterozygous in a population if more than half individuals were heterozygous in that population but not in the other population. Variations in exons were further categorized as synonymous or non-synonymous SNPs. Two-sided two-sample Wilcoxon signed-rank tests were conducted to test whether the ratios of genes with nonsynonymous mutations were different between northern and southern populations, using the function *wilcoxsign-test* in R package "coin".

**Population analysis**. Population structure was inferred using ADMIXTURE v.1.23[107] with default settings. The number of assumed genetic clusters $K$ ranged from 2 to 5, and the optimum $K$ was assessed with cross-validation errors. The individual-based Maximum Likelihood (ML) phylogenetic tree was constructed using MEGA[108] under Jukes-Cantor model with 1000 bootstraps, and visualized using FigTree. PCA was performed with whole-genome SNPs of all 264 individuals using Eigensoft[109]. To evaluate LD decay, parameter $r^2$ between any two loci was calculated within each chromosome using Plink v.1.07[106] with the command –ld-window-r2 0 –ld-window 99999 –ld-window-kb 500. The average $r^2$ values were calculated for each length of distance and the genome-wide LD was averaged across all chromosomes. The LD decay was plotted against the length of distance. Popgenome R package[110] was used to calculate Tajima's $D$, global $F_{ST}$ and nucleotide diversity ($p$) using a 100-kb sliding window with the step size of 10-kb.

**Demographic history**. We implemented PSMC[111] to estimate dynamics of effective population size ($Ne$) and possible divergence time over the past several million years. A total of eight estuarine oysters from northern (n = 3), middle (n = 2) and southern (n = 3) populations and three Pacific oysters[12] with high sequencing depth (20 − 28×) were used. To minimize the probability of false positives, sequencing depth of SNPs was filtered with parameters: MinDepth = average depth/3, MaxDepth = average depth×2. The PSMC parameters were set as: -N25 -t15 -r5 -p '4 + 25 * 2 + 4 + 6' to estimate historical $Ne$. The estimated generation time ($g$) was set as 1 for both species, while mutation rates ($\mu$) were calculated, following the formula $T_{\_divergence} = Ks/2\mu$, as $0.3×10^{-8}$ and $0.2×10^{-8}$ for estuarine and Pacific oysters, respectively.

**Selection signature detection**. To identify selection sweeps potentially contributing to adaptation to southern environments, we calculated fixation ($F_{ST}$) and selection statistics (Tajima's $D$) between two pairs of populations, north vs. south and middle vs. south, in a 100-kb sliding window with a step size of 10-kb. Genomic regions showing strong selection signals were defined as: (1) regions with top 1% $F_{ST}$ values that overlapped in both pair comparisons; and (2) regions located at valleys of Tajima's $D$ values distribution along the chromosome in one of the three populations.

**Expansion of candidate genes under selection**. Orthologs between genes from four oyster species with high-quality genomes, *C. ariakensis*, *C. hongkongensis*[33], *C. virginica* (NCBI assembly Bioproject PRJNA376014, GCA_002022765.4 C_virginica-3.0) and *C. gigas*[34], were determined with Orthofinder[112] v2.3.7, with default parameters using protein sequences (e-value = $1e^{-5}$). For expanded *Slc* gene families under selection, we constructed phylogenetic tree for orthologs belong to *Slc23a2* and *Mct12* gene families in *Crassostrea* oysters, including *C. gigas*, *C. ariakensis*, *C. hongkongensis* and *C. virginica*. The phylogenetic relationship was inferred with protein sequences using the Maximum Likelihood (ML) method and the Whelan And Goldman (WAG) model in MEGA7 software[108], with 1,000 bootstraps.

**Response to temperature and salinity disturbance**. To investigate responses of estuarine oysters to challenges of elevated temperature and high salinity, we collected wild oysters and exposed them to different temperatures (20 °C and 37 °C for 6 h) and salinities (20 ‰ and 60 ‰ for 7 days), respectively. The control oysters were exposed to seawater of 20 ‰ salinity at 20 °C. Gills from five oysters were individually sampled and flash-frozen in liquid nitrogen for subsequent transcriptomic analysis.

**Reciprocal transplantation experiments**. To explore the plasticity of candidate genes from regions under selection, we measured transcriptional changes in oysters subjected to native and non-native environments after reciprocal transplantation[3]. Briefly, wild oysters derived from northern (Binzhou: BZ, Bohai Sea) and southern (Taishan: TS, East China Sea) environments were collected and used to produce $F_1$ progeny. For each population, 40 males and 40 females were selected as parents. Eggs were pooled and then divided into 40 beakers, each fertilized with sperm from one of the 40 males to create all possible crosses. Zygotes fathered by every eight males were combined and cultured as one group, and the five groups were reared in separate tanks in the hatchery and nursery to juvenile stage. Two-month-old $F_1$ juveniles from each of the two populations were outplanted at two source habitats to assess their responses to reciprocal transplantation or environmental change. After three months of acclimation at northern and southern environments, gills of five oysters were sampled from each of population at both habitats in situ and flash-frozen in liquid nitrogen for subsequent transcriptomic analysis.

**Transcriptomic analysis**. Total RNA was isolated from gills sampled from laboratory challenge experiment (high temperature and high salinity) and field reciprocal transplantation experiment, using the RNAprep Pure Tissue Kit (Tiangen) following the manufacturer's protocol. The RNA integrity and concentration were examined with 1.2% agarose gel electrophoresis and Nanodrop 2000 spectrophotometer, respectively. DNA contamination was removed with DNAse I treatment. RNA integrity was further assessed using an Agilent Bioanalyzer 2100 and the RNA Nano 6000 Assay Kit. One μg RNA per sample was used to construct sequencing libraries using NEBNext Ultra™ RNA Library Prep Kit for 150-bp paired-end sequencing on an Illumina HiSeq 4000 platform. Clean data were obtained by removing reads containing adapters, reads containing ploy-N and low-quality reads. TopHat2 was used to map clean reads to the estuarine oyster genome. StringTie v2.0 was used for read assembly. Only reads with perfect match or one mismatch were further analyzed and annotated. Gene expression levels were estimated by fragments per kilobase of transcript per million fragments mapped (FPKM). We used DESeq2 to determine differentially expressed genes (DEGs) between different populations or treatments. Genes with an adjusted $p$-value < 0.01 using the Benjamin and Hochberg's approach were accepted as DEGs. For genes from regions under selection, those significantly differentially expressed between northern and southern habitats in both oyster populations were defined as high plasticity genes (HP), while others were defined as low plasticity genes (LP). A hierarchical cluster analysis was performed to show differential expression of these genes, using the *pheatmap* package in R software.

**Statistics and reproducibility**. The difference of gene expression level between northern and southern oyster populations at two environmental habitats and those under thermal and high-salinity stressed conditions were determined by DESeq2 in R software. A total of 264 wild oysters collected from 11 estuaries were used for whole-genome resequencing and 40 individuals were used to compared gene expression difference under different habitats, thermal and high-salinity conditions. For all figures, the criterion for statistical significance was set as *$P < 0.05$, **$P < 0.01$ and ***$P < 0.001$.

**Reporting summary**. Further information on research design is available in the Nature Research Reporting Summary linked to this article.

## Data availability

The genome, whole-genome re-sequencing and transcriptome datasets were deposited in the Sequence Read Archive (SRA) database under the accession number PRJNA715058, or GenBank under the accession number JAGFMG000000000.1. In addition, we uploaded annotation files of estuarine oyster genome, including gene locations (.gff3), protein models (.pep), CDS (.cds) and exon (.exon) sequences to FigShare online (https://figshare.com/articles/dataset/Genome_of_the_estuarine_oyster_provides_insights_into_climate_impact_and_adaptive_plasticity/16557390). Source data underlying Figs. 3 and 5b, c were provided in Supplementary Data 7.

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

## Acknowledgements

We thank X. Wang, Z. Jia, Z. She, Y. Zhang, Z. Yu, W. Quan, Z. Huo, X. Yan, Z. Zeng and Y. Ning for sample collection, and B. Yin and J. Qi for information on marine currents. L.L. is supported by the Strategic Priority Research Program of the Chinese Academy of Sciences (No. XDA23050402) and the National Key R&D Program of China (No. 2018YFD0900304). A.L. is supported by the National Natural Science Foundation of China (No. 32101353) and by China Postdoctoral Science Foundation (No. 2019TQ0324). A.L. and L.L. are supported by Key Deployment Project of Centre for Ocean Mega-Research of Science, Chinese Academy of Sciences (No. COMS2019Q06). A.L. is also supported by the Distinguished Young Scientists Research Fund of Key Laboratory of Experimental Marine Biology, Chinese Academy of Sciences (No. KLEMB-DYS04). L.L. is also supported by the Technology and the Modern Agro-industry Technology Research System (No. CARS-49).

## Author contributions

L.L., G.Z. and X.G. conceived the study and participated in final data analysis, inter-pretation and manuscript writing. A.L. carried out data analysis and drafted the manuscript. H.D., A.L., H.C., X.L. and H.Z. contributed to the selective sweep analysis. A.L., Z.Z., K.Z., C.W. and X.W. collected and sampled oyster specimens. A.L. and W.W. produced the F$_1$ progeny. A.L., L.L., X.G. and G.Z. revised the manuscript. All authors approved the manuscript for publication.

## Competing interests

The authors declare no competing interests.
