## [Transparent Peer Review File · Communications Biology]

Reviewers' comments:

Reviewer #1 (Remarks to the Author):

The authors present a high quality genome from *C. ariakensis*, and analyses on the selective pressures and population structure in this species. The genome assembly is remarkable (200x long reads and low heterozygosity helped), and the analyses of population structure and evolutionary history are compelling and backed up with geological and oceanographic evidence. The authors highlight a suite of genes likely important for adaptation to environments of different salinities, and the reciprocal transplant experiments and subsequent gene expression studies provide evidence for this hypothesis. The descriptions of the analyses and statistical follow-ups are straightforward throughout the paper. On the whole the manuscript is interesting, and this genome will be of-use to other researchers. With minor revision, I think the manuscript can proceed toward publication.

Required changes:

Line 98: Your genome is better than just "the most contiguous bivalve"! Compare quality/completeness to recent mollusc genomes (Scaly-foot snail, Fuzzy chiton, Bobtail squid, maybe even the recent *Helobdella*, etc.)- this work will be useful for many broader studies.

Line 309: It would be helpful to provide phylogenetic evidence for the gene expansions listed here by building a tree with the relevant orthogroups across the species used to generate orthogroups. For example, I would expect a clade of *Slc23a2* that contains multiple species, and a subclade of *C. ariakensis* with additional tips that are more closely related to one another than to the other species or to other solute carriers. Basically, phylogenetic evidence would help to support that these purported expansions are genuine biological characters and not errors in assembly, etc. Additionally, expression data for all copies, not just those that responded to challenges, should be available in a supplementary table to discount the idea that this could just be a plethora of pseudogenes. This is a big part of what makes your paper better than just "we sequenced a genome", so it would be helpful to back up the copy number section with more evidence. Especially because it's followed by such an elegant piece of evidence (the differential gene expression in response to environment).

Line 321: I may have missed it, but to my eyes no methods are presented for the generation of orthologs. Which taxa were included in the initial orthogroup generation, what software was used, and what parameters were used for clustering, etc.? How did you decide which orthogroups were 'valid'? Was this an analysis of only *Crassostrea*, or are other molluscs included for comparison?

Line 587: This is nitpicky, but PRJ is your project number. It may be helpful to provide a table in your supplemental data of all the accession numbers and what they correlate to, especially because this study encompassed one sequencing individual and then several other conspecifics. You should have a genome assembly number and separately SRA numbers for all raw read data sets. Also, please specify if the annotation produced in this manuscript was uploaded to GenBank. If not, please provide the protein models online elsewhere (FigShare, etc.).

Line 593: What method was used to extract genomic DNA?

Line 593: Which library chemistry, what flow-cell type, and what machine was used to generate Nanopore sequencing?

Line 610: Please specify the number of changes in ideally all rounds but absolutely in the final round of correction by both Racon and Pilon. Alternatively, the output files from each round could be made available online.

Line 638: I am surprised that Augustus is used here, untrained for the taxon of interest. Please state

what guide species was selected for Augustus. (Default is fly?)

Line 640: hard or soft-masking for each analysis?

Line 641: The taxon sampling here is a bit odd; there are several other high-quality molluscan genomes available that are not used here, and a few listed that are known to have questionable annotations. Clearly this group worked, but some explanation of the selection may be in order.

Line 647: What method was used to extract RNA, and again what library preparation kit, flowcell, and instrument was used to generate the sequence?

Line 652: The weights file should either be available in the supplement, or weights can be listed in text here.

Other edits and suggestions:

It was disappointing to not see a comparison of synteny between this genome and one of the other high-quality *Crassostreas* (aka *virginica*). Would take ~2 hours with SynMap2 and could be interesting just to visualize.

Line 101: I would put the specific BUSCO gene set used in both the Results and the Methods (it says version 3.0.2 but is this obd9? It may be worth re-analyzing with obd10 – it's a better gene set (and honestly tends to bump mollusc scores up 0.5% or so!).

Line 103: The accuracy measurement here is fine but is not typical enough that it can stand as a result without specifying the method. I would change to "The accuracy of genome sequence, as quantified by the percentage of RNA-seq reads and Illumina genomic reads". Did you use a tool (e.g REAPR) to accomplish this? If time permits, REAPR takes a few hours to run on an existing alignment and can add additional support to the accuracy of your assembly by looking specifically for errors in scaffolding; I doubt it will find anything in your excellent assembly but it's a nice way of quantifying this "accuracy" with a known tool.

Edits for clarity:

Line 37: "most of the energy" – meaning unclear

Line 70: "only found in estuaries at low abundance within its wide distribution range" – if correct, rephrase to "only found in lower-salinity estuaries that are less abundant habitats within its wide distribution range".

Reviewer #2 (Remarks to the Author):

The manuscript by Li et al. aims at exploring the genomic variation and population structure in the estuarine oyster *Crassostrea ariakensis*. For that, the authors generated a chromosome-level genome assembly for the species together with whole-genome resequencing data for more than 260 individuals from several populations covering its latitudinal distribution. Overall, this is a well-designed, well-executed and well-written study of potential high interest for a wide readership interested in evolutionary biology in general and biodiversity genomics in particular. It is also a sound case study of how chromosome-level genomes can be leveraged to deepen our understanding of particular biological and evolutionary questions.

I only have a few comments and suggestions that would like to be addressed.

One of my main concerns is the use of neighbor joining to build the phylogenetic tree with SNP data as reported in Fig. 2c. The main problem with the neighbor-joining scheme is that in the first step, the distances are estimated from noisy data, and consequently, the resulting dissimilarity map is very unlikely to be a tree metric. Neighbor joining has been largely superseded by phylogenetic methods that do not rely on distance measures and offer superior accuracy under most conditions. A maximum likelihood or Bayesian inference tree would be more adequate and would result in a more reliable inference.

A second concern is related to the methods. It is difficult to understand what was done exactly and why. I'd like to see more details about the methodology. For example, how was high molecular weight DNA extracted? How many flow cells were run for Oxford Nanopore sequencing? Same for the RNAseq data (see line 647). Where did the sequencing take place? Why 21-mer for KCN analyses? What other kmer size tested? Was the genome size estimation by kmer analysis concordant with the final assembly size? How was the HiC library constructed (kits, conditions, etc)? And how much data was generated for the HiC library? Line 730: what do the 20 and 60 numbers mean in this sentence? Please elaborate on this paragraph a bit more. Where control oysters used for the gene expression analysis?

Other very minor comments:

Fig. 1. Figure 1e is missing (duplicated gene cluster of 123 the solute carrier families showing selection signals).

Line 284: . Only genes 285 with 10 or more aligned reads in >90% samples were used for subsequent analyses -> what is the reason for this? Differential gene expression analyses are designed to take into account these factors if replicates are included in the study.

Line 289: delete 'and' in front of 75.9%

Line 400: I'd suggest rephrasing the sentence as follows: 'Expansion and selection in upstream regulatory regions of environment-responsive genes may hence be critical for the adaptation to rapidly changing environments in the estuarine oyster.' The part of enhancing the transcriptional complexity and phenotypic plasticity is not very thoroughly tested in this manuscript, although the results point to it.

Data availability: it is mentioned that the genome, whole genome resequencing and transcriptomic data are deposited in the cited BioProject. I strongly encourage the authors to deposit as well the genome annotations, since the gene repertoire is also discussed in the manuscript.

Reviewer #1 (Remarks to the Author):

Required changes:

Line 98: Your genome is better than just "the most contiguous bivalve"! Compare quality/completeness to recent mollusc genomes (Scaly-foot snail, Fuzzy chiton, Bobtail squid, maybe even the recent *Helobdella*, etc.)- this work will be useful for many broader studies.

>>>Response: We appreciate the positive assessment and added a supplementary stable comparing quality/completeness of recent mollusc genomes.

“To our knowledge, this is the most contiguous assembly (by contig N50) produced with nanopore long-read sequencing for a bivalve mollusc²⁹⁻³⁸ (Supplementary Table 3).” (Lines 100-102)

Supplementary Table 3. Comparison of the quality among mollusc genomes.

Species	Genome size (Mb)	Gene number	Contig N50 (Mb)	Scaffold N50 (Mb)	BUSCO (%)
Crassostrea ariakensis	613.9	29,631	7.0	62.3	92.2
Crassostrea hongkongensis	610.0	25,675	2.6	55.6	95.8
Crassostrea virginica	684.7	39,493	2.0	75.9	94.5
Crassostrea gigas^a	647.9	30,724	1.8	58.4	95.6
Crassostrea gigas^b	586.8	30,078	3.1	60.9	92.5
Chrysomallon squamiferum	444.4	16,917	1.9	30.2	96.6
Acanthopleura granulata	606.9	20,470	1.1	23.9	97.4
Mercenaria mercenaria	1,780	34,283	1.8	91.4	90.5
Scapharca broughtonii	884.5	24,045	1.8	45.0	91.7
Tegillarca granosa	812.6	24,398	0.6	42.6	93.3
Nautilus pompilius	730.6	17,710	1.1	/	93.5
Euprymna scolopes	5,280	29,259	0.0036	3.7	97.0

Helobdella					
robusta	235.4	23,400	0.052	3.1	/

a: assembled by Penalzoza, et al. (2021). b: assembled by Qi, et al. (2021).

Line 309: It would be helpful to provide phylogenetic evidence for the gene expansions listed here by building a tree with the relevant orthogroups across the species used to generate orthogroups. For example, I would expect a clade of *Slc23a2* that contains multiple species, and a subclade of *C. ariakensis* with additional tips that are more closely related to one another than to the other species or to other solute carriers. Basically, phylogenetic evidence would help to support that these purported expansions are genuine biological characters and not errors in assembly, etc. Additionally, expression data for all copies, not just those that responded to challenges, should be available in a supplementary table to discount the idea that this could just be a plethora of pseudogenes. This is a big part of what makes your paper better than just "we sequenced a genome", so it would be helpful to back up the copy number section with more evidence. Especially because it's followed by such an elegant piece of evidence (the differential gene expression in response to environment).

>>>Response: We agree and appreciate the suggestion. We added phylogenetic analysis and supplementary table of expression data for all expanded gene copies in the revised MS.

For phylogenetic analysis:

“In addition to their presence in tandem arrays on chromosome 9, the expansion of the *solute carrier* families was further supported by phylogenetic analysis where duplicated genes were mostly clustered in orthologroup- and lineage/species-specific manners (Supplementary Fig. 17).” (Lines 348-352)

“For expanded *Slc* gene families under selection, we constructed phylogenetic tree for orthologs belong to *Slc23a2* and *Mct12* gene families in *Crassostrea* oysters, including *C. gigas*, *C. ariakensis*, *C. hongkongensis* and *C. virginica*. The phylogenetic relationship was inferred with protein sequences using the Maximum Likelihood (ML) method and the Whelan And Goldman (WAG) model in MEGA7 software¹⁰⁸, with 1,000 bootstraps.” (Lines 813-818)

For expression data:

We provided expression data for all expanded *Slc23a2* and *Mct12* genes from chromosome 9 in Supplementary Table 14 with the following text.

“Gene copies that did not respond to heat and high-salinity stresses were not expressed or expressed at very lower expression (FPKM < 0.5) (Supplementary Table 14). Conversely, three copies of *Slc23a2* (*Slc23a2_a1*, *Slc23a2_a2* and *Slc23a2_c2*) in two expanded orthogroups responded to both temperature and salinity challenges, while three copies of *Mct12* (*Mct12_1*, *Mct12_2* and *Mct12_3*) responded to salinity challenge only (Fig. 4c) and preferentially clustered together (*Mct12_1/2/3* and *Slc23a2_a1/2*, Supplementary Fig. 17).” (Lines 352-358)

Supplementary Fig. 17 | Phylogenetic tree of orthologs belong to *Slc23a2* and *Mct12* gene families in *Crassostrea* oysters inferred by the Maximum Likelihood (ML) method with 1,000 bootstraps. Green branches indicate most of orthologs (14 of 17) belong to orthogroup OG0000489, purple branches indicate orthologs belong to orthogroups OG0011985 and OG0000633, and blue branches indicate orthologs belong to orthogroup OG0000571. Car: *C. ariakensis* (purple), Cho: *C. hongkongensis* (blue), Cgi: *C. gigas* (black), Cvi: *C. virginica* (green).

Line 321: I may have missed it, but to my eyes no methods are presented for the generation of orthologs. Which taxa were included in the initial orthogroup generation, what software was used, and what parameters were used for clustering, etc.? How did you decide which orthogroups were ‘valid’? Was this an analysis of only *Crassostrea*, or are other molluscs included for comparison?

>>>Response: Sorry for the missing details which were added to **Methods** in the revised MS.

“Orthologs between genes from four oyster species with high-quality genomes, *C. ariakensis*, *C. hongkongensis*³³, *C. virginica* (NCBI assembly Bioproject PRJNA376014, GCA_002022765.4 C_virginica-3.0) and *C. gigas*³⁴, were determined with Orthofinder¹¹² v2.3.7, with default parameters using protein sequences (e-value = 1e⁻⁵.” (Lines 809-813)

33 Peng, J. *et al.* Chromosome-level analysis of the *Crassostrea hongkongensis* genome reveals extensive duplication of immune-related genes in bivalves. *Molecular ecology resources* **20**, 980-994, doi:10.1111/1755-0998.13157 (2020).

34 Qi, H., Li, L. & Zhang, G. Construction of a chromosome-level genome and variation map for the Pacific oyster *Crassostrea gigas*. *Molecular ecology resources*, doi:10.1111/1755-0998.13368 (2021).

112 Emms, D. M. & Kelly, S. OrthoFinder: phylogenetic orthology inference for comparative genomics. *Genome biology* **20**, 238, doi:10.1186/s13059-019-1832-y (2019).

Line 587: This is nitpicky, but PRJ is your project number. It may be helpful to provide a table in your supplemental data of all the accession numbers and what they correlate to, especially because this study encompassed one sequencing individual and then several other conspecifics. You should have a genome assembly number and separately SRA numbers for all raw read data sets. Also, please specify if the annotation produced in this manuscript was uploaded to GenBank. If not, please provide the protein models online elsewhere (FigShare, etc.).

>>>Response: We understand the point. We provided a Supplementary Table 18 for all accession numbers (including genome assembly number and SRA numbers for all raw

read data sets) and the information of each oyster individuals sampling location, genetic group and treatment. In addition, we uploaded the details for genome annotation of estuarine oyster, including gene locations (.gff3), protein models (.pep), CDS (.cds) and exon (.exon) sequences in FigShare online.

(https://figshare.com/articles/dataset/Genome_of_the_estuarine_oyster_provides_insights_into_climate_impact_and_adaptive_plasticity/16557390).

Line 593: What method was used to extract genomic DNA?

>>>Response: Sorry for the confusion. We extracted genomic DNA using DNeasy Blood & Tissue Kit (Qiagen, Hilden, Germany). We added this detail in the revised MS: “Genomic DNA was extracted from the adductor muscle using the DNeasy Blood & Tissue Kit (Qiagen, Hilden, Germany)”. (Lines 634-635)

Line 593: Which library chemistry, what flow-cell type, and what machine was used to generate Nanopore sequencing?

>>>Response: We added the following details in the revised MS.

“gDNA (2µg) was repaired using NEB Next FFPE DNA Repair Mix kit (M6630, USA) and subsequently processed using the ONT Template prep kit (SQK-LSK109, UK) according to the manufacturer’s instructions. The large segment library was premixed with loading beads and then pipetted into a previously used and washed R9 flow cell. The library was sequenced on the ONT PromethION platform with nine R9 cells and ONT sequencing reagent kit (EXP-FLP001.PRO.6, UK) according to the manufacturer’s instructions.” (Lines 636-642)

Line 610: Please specify the number of changes in ideally all rounds but absolutely in the final round of correction by both Racon and Pilon. Alternatively, the output files from each round could be made available online.

>>>Response: We provided the number changes of all rounds of Pilon correction (no data for Racon correction), and the final round of correction by both Racon and Pilon in the revised MS as Supplementary Table 15 and 16.

“Quickmerge⁷¹ v0.2.2 was used to join the three assemblies, and the assembly was corrected for 3 cycles with long reads using Racon⁷² and for 3 cycles with Illumina reads using Pilon⁷³ v1.22 with default parameters (Supplementary Table 15 and 16). Then, the Purge Haplotigs⁷⁴ software was used to remove redundant heterozygous contigs with parameter ‘purge_haplotigs purge -a 55’.” (Lines 662-667)

Supplementary Table 15 | Summary for three rounds of correction by Pilon.

Genome_size	SNP	Insert	Delete	Error ratio (%)
777,213,952	442,861	393,016	2,378,169	0.41
779,067,884	98,576	26,236	59,702	0.02
779,041,778	29,647	6,240	12,946	0.01

Supplementary Table 16 | Summary for the final round of correction by both Racon and Pilon.

Correction	Contig Number	Genome size	Contig N50	Contig N90	Contig Maximum	GC(%)
Before Racon	1,777	773,515,823	5,384,006	270,489	20,131,303	33.36
After Racon	1,777	777,213,952	5,454,258	274,035	20,452,959	33.34
After Pilon	1,777	779,025,439	5,465,454	274,941	20,491,775	33.39

74 Roach, M. J., Schmidt, S. A. & Borneman, A. R. Purge Haplotigs: allelic contig reassignment for third-gen diploid genome assemblies. *Bmc Bioinformatics* **19**, 460, doi:10.1186/s12859-018-2485-7 (2018).

Line 638: I am surprised that Augustus is used here, untrained for the taxon of interest.

Please state what guide species was selected for Augustus. (Default is fly?)

>>>Response: Sorry for the confusion. The transcriptomic data of the estuarine oyster was selected for training of Augustus. We corrected this sentence as: “For *de novo* prediction, five *ab initio* gene prediction programs, Genscan⁸⁰ v1.0, Augustus⁸¹ v2.4 (with transcriptomic data of the estuarine oyster used for training), GlimmerHMM⁸² v3.0.4, GeneID⁸³ v1.4 and SNAP⁸⁴, were used to predict genes in the repeat-masked genome (hard-masking).” (Lines 701-704)

Line 640: hard or soft-masking for each analysis?

>>>Response: We used hard-masking for each analysis. We added this information in the revised MS: “For *de novo* prediction, five *ab initio* gene prediction programs,

Genscan⁸⁰ v1.0, Augustus⁸¹ v2.4 (with transcriptomic data of the estuarine oyster used for training), GlimmerHMM⁸² v3.0.4, GeneID⁸³ v1.4 and SNAP⁸⁴, were used to predict genes in the repeat-masked genome (hard-masking).” (Lines 701-704)

Line 641: The taxon sampling here is a bit odd; there are several other high-quality molluscan genomes available that are not used here, and a few listed that are known to have questionable annotations. Clearly this group worked, but some explanation of the selection may be in order.

>>>Response: We agree and provided explanation for taxon selection in the revised MS.

“For homolog-based prediction, protein sequences from 10 well-annotated species that were annotated by the NCBI Eukaryotic Genome Annotation Pipeline, *Homo sapiens*, *Danio rerio*, *Aplysia californica*, *Strongylocentrotus purpuratus*, *C. gigas*, *C. virginica*, *Biomphalaria glabrata*, *Lingula anatina*, *Octopus bimaculoides* and *Mizuhopecten yessoensis*,” (Lines 704-708)

Line 647: What method was used to extract RNA, and again what library preparation kit, flowcell, and instrument was used to generate the sequence?

>>>Response: Sorry for the confusion. We provided the following details in the revised MS.

“four tissues (gill, mantle, adductor muscle and labial palp) were collected from the same oyster used for genome sequencing. Total RNA was extracted separately using TRIzol reagent (OMEGA, USA) according to the manufacturer’s instructions. RNA quality and quantity were assessed using the Agilent 2100 instrument and Qubit 2.0 fluorometer (Thermo Fisher, USA), respectively. RNA from four tissues were combined in equimolar quantities into a single pool. Full-length coding DNA (cDNA) was obtained using SMARTer PCR cDNA Synthesis Kit (Clontech, USA). The library was constructed using SMRTbell[®] Template Prep Kit 2.0 (PacBio, USA). cDNA was sequenced by PacBio Sequel II platform with SMRT cell 8M (Biomarker, China).” (Lines 711-720)

Line 652: The weights file should either be available in the supplement, or weights can be listed in text here.

>>>Response: Sorry for the confusion. We added weight values for each method of gene prediction as Supplementary Table 17 in the revised MS.

Supplementary Table 17. Weight value for gene prediction indicated by homology, *de novo* prediction and mRNA transcripts.

Method	Software	Weight value
Homology-based	GeMoMa	50
RNAseq	PASA	50
Ab initio	Genscan	0.3
	AUGUSTUS	0.3
	SNAP	0.3
	GeneID	0.3
	GlimmerHMM	0.3

Other edits and suggestions:

It was disappointing to not see a comparison of synteny between this genome and one of the other high-quality *Crassostreas* (aka *virginica*). Would take ~2 hours with SynMap2 and could be interesting just to visualize.

>>>Response: We appreciate the point and compared syntenic relationships among three oyster species, *C. virginica*, *C. hongkongensis* and *C. ariakensis*, that inhabit low-salinity estuarine. We added the following to **Results** and **Methods** in the revised MS. “We analyzed macrosynteny between *C. ariakensis* and two other oyster species inhabiting low-salinity estuarine, *C. virginica* and *C. hongkongensis*. High colinearity was found between *C. ariakensis* and *C. hongkongensis* across 205.48 Mb covering 20,571 genes of *C. ariakensis* genome (Supplementary Table 9), while a lower colinearity between *C. ariakensis* and *C. virginica* was detected across 194.17 Mb covering 18,692 genes of *C. ariakensis* genome (Supplementary Fig. 3, Supplementary Table 10). Our findings are consistent with the fact that the two Asian species *C. ariakensis* and *C. hongkongensis* have a closer phylogenetic relationship that diverged 22.3 Myr ago, while the Atlantic *C. virginica* diverged from the Asian species about 82.7 Myr ago³⁹. Assembly errors in the *C. virginica* genome as indicated by the

discrepancy with linkage map (X. Guo, personal communication) may also explain some of the differences.” (Lines 124-135)

“We analyzed macrosynteny between *C. ariakensis* and the other two oyster species living in low-salinity estuaries, *C. virginica* and *C. hongkongensis*. First, we aligned gene sequences between two oyster species using Diamond¹⁰¹ v0.9.29.130, to identify homologous gene pairs (e-value < 1e⁻⁵, C score > 0.5). MCSanX¹⁰² was used to analyze chromosome collinearity for syntenic blocks between *C. ariakensis* and the other two oyster species (alignment significance = 1e⁻¹⁰), using Diamond result file and gff file. The maximum gap size was set to 25 genes and a minimum syntenic block required 5 genes.” (Lines 738-745)

Supplementary Fig. 3 | Comparison of syntenic relationships between *C. ariakensis* and other two oyster species living in low-salinity estuaries, *C. hongkongensis* and *C. virginica*.

39 Ren, J., Liu, X., Jiang, F., Guo, X. & Liu, B. Unusual conservation of mitochondrial gene order in *Crassostrea* oysters: evidence for recent speciation in Asia. *BMC evolutionary biology* **10**, 394, doi:10.1186/1471-2148-10-394 (2010).

101 Buchfink, B., Xie, C. & Huson, D. H. Fast and sensitive protein alignment using DIAMOND. *Nat Methods* **12**, 59-60, doi:10.1038/nmeth.3176 (2015).

102 Wang, Y. *et al.* MCSanX: a toolkit for detection and evolutionary analysis of gene syteny and collinearity. *Nucleic acids research* **40**, e49, doi:10.1093/nar/gkr1293 (2012).

Line 101: I would put the specific BUSCO gene set used in both the Results and the Methods (it says version 3.0.2 but is this obd9? It may be worth re-analyzing with obd10 – it’s a better gene set (and honestly tends to bump mollusc scores up 0.5% or so!).

>>>Response: We appreciate the point and re-analyzed with obd10. The assembly of estuarine oyster captured 92.24% of the BUSCO datasets. We put the specific BUSCO gene set in both **Results** and **Methods** as Supplementary Table 5.

“Also, the genome assembly captured 92.24% of the Benchmarking Universal Single Copy Orthologs (BUSCO) datasets (Fig. 1d, Supplementary Table 5),” (Lines 105-107)

“Benchmarking Universal Single-Copy Orthologs (BUSCO, obd10) with 954 conserved genes were used to assess the completeness and accuracy of gene coverage.” (Lines 684-685)

Line 103: The accuracy measurement here is fine but is not typical enough that it can stand as a result without specifying the method. I would change to “The accuracy of genome sequence, as quantified by the percentage of RNA-seq reads and Illumina genomic reads”. Did you use a tool (e.g REAPR) to accomplish this? If time permits, REAPR takes a few hours to run on an existing alignment and can add additional support to the accuracy of your assembly by looking specifically for errors in scaffolding; I doubt it will find anything in your excellent assembly but it’s a nice way of quantifying this “accuracy” with a known tool.

>>>Response: We agree and corrected this sentence as: “The coverage of the assembled genome was assessed by mapping RNA-seq reads and Illumina genomic reads, and over 97.9% of genomic short-reads and 97.2% RNA-seq reads were mapped to the assembly respectively (Supplementary Table 4).” (Lines 102-105)

In addition, we provided the results of REAPR in the **Methods** and **Results** of revised MS.

“Further, the accuracy of genome sequences was assessed by REAPR⁷⁷ v. 1.0.18 with default parameters to identify errors in genome assemblies.” (Lines 688-690)

“Analysis with REAPR showed that both fragment coverage distribution error and low fragment coverage over a gap were zero, indicating that the genome assembly was accurate.” (Lines 109-111)

⁷⁷ Hunt, M. *et al.* REAPR: a universal tool for genome assembly evaluation. *Genome biology* **14**,

R47 (2013).

Edits for clarity:

Line 37: “most of the energy” – meaning unclear

>>>Response: Sorry for the confusion. We corrected it as: “Oceans bear the blunt of climate change as they absorb most of the heat energy from the sun”. (Lines 37-38)

Line 70: “only found in estuaries at low abundance within its wide distribution range” – if correct, rephrase to “only found in lower-salinity estuaries that are less abundant habitats within its wide distribution range”.

>>>Response: Thanks for pointing out the confusion. We corrected it as “the estuarine oyster is only found in lower-salinity estuaries within its wide distribution range and at low abundance.” (Lines 70-71)

Reviewer #2 (Remarks to the Author):

I only have a few comments and suggestions that would like to be addressed.

One of my main concerns is the use of neighbor joining to build the phylogenetic tree with SNP data as reported in Fig. 2c. The main problem with the neighbor-joining scheme is that in the first step, the distances are estimated from noisy data, and consequently, the resulting dissimilarity map is very unlikely to be a tree metric. Neighbor joining has been largely superseded by phylogenetic methods that do not rely on distance measures and offer superior accuracy under most conditions. A maximum likelihood or Bayesian inference tree would be more adequate and would result in a more reliable inference.

>>>Response: We agree and replaced the phylogenetic tree in the revised MS using maximum likelihood, which was consistent with the results using neighbor joining (Fig. S1). We also revised the corresponding sentence in the revised MS: “Moreover, phylogenetic analysis using the Maximum Likelihood (ML) method supported PCA clustering, by first distinguishing southern populations from others and then identified a subpopulation (NC-a) within NC located in estuaries of southern Bohai Sea including

Binzhou (BZ) and Dongying (DY) (Fig. 2c).” (Lines 163-167)

“The individual-based Maximum Likelihood (ML) phylogenetic tree was constructed using MEGA¹⁰⁸ under Jukes-Cantor model with 1000 bootstraps, and visualized using FigTree.” (Lines 778-780)

Fig. S1 Phylogenetic tree of estuarine oysters inferred from genome-wide SNPs by (a) the neighbour-joining (NJ) and (b) the maximum likelihood (ML) methods. NC-a: northern China, including BZ and DY; NC-b: northern China including DD and YK; MC: middle China including NT and SH; SC-a: southern China including JLJ and ZhJ; and SC-b: southern China including TS and QZh.

Fig. 2 | Geographic distribution and genetic differentiation of the estuarine oyster populations. **a**, Sampling locations of 264 resequenced wild oysters from 11 estuaries along the coast of China. The arrowed curves represent ocean currents in summer: a: Kuroshio Current, b: Yellow Sea Warm Current, c: Yellow Sea Cold Water Mass, d: Bohai Sea Circulation, e: China Coastal Current, f: South China Sea Warm Current. SCS: South China Sea, ECS: East China Sea, YS: Yellow Sea, BS: Bohai Sea. DD: Dandong, YK: Yingkou, BZ: Binzhou, DY: Dongying, QD: Qingdao, NT: Nantong, SH: Shanghai, JLJ: Jiulongjiang, ZhJ: Zhangjiang, TS: Taishan, QZh: Qinzhou. **b**, Plots of principal components 1 and 2 of 264 resequenced oysters based on whole-genome data. **c**, Phylogenetic tree of estuarine oysters inferred from genome-wide SNPs with the maximum likelihood (ML) method. NC-a: northern China, including BZ and DY; NC-b: northern China including DD and YK; MC: middle China including NT and SH; SC-a: southern China including JLJ and ZhJ; and SC-b: southern China including TS and QZh. **d**, Nucleotide diversity (p , under or above the circles) and genetic divergence (F_{ST} , between populations) among four populations.

A second concern is related to the methods. It is difficult to understand what was done exactly and why. I'd like to see more details about the methodology. For example, how

was high molecular weight DNA extracted? How many flow cells were run for Oxford Nanopore sequencing? Same for the RNAseq data (see line 647). Where did the sequencing take place? Why 21-mer for KCN analyses? Was other kmer size tested? Was the genome size estimation by kmer analysis concordant with the final assembly size? How was the HiC library constructed (kits, conditions, etc)? And how much data was generated for the HiC library? Line 730: what do the 20 and 60 numbers mean in this sentence? Please elaborate on this paragraph a bit more. Where control oysters used for the gene expression analysis?

>>>Response: Sorry for the confusion. We provided these details in the revised MS.

“Genomic DNA was extracted from the adductor muscle using the DNeasy Blood & Tissue Kit (Qiagen, Hilden, Germany)” (Lines 634-635)

Nanopore long-reads were generated from nine Oxford Nanopore flowcells on the PromethION platform (183.70 Gb, 299.24×). “The library was sequenced on the ONT PromethION platform with nine R9 cells and ONT sequencing reagent kit (EXP-FLP001.PRO.6, UK) according to the manufacturer’s instructions.” (Lines 640-642)

“four tissues (gill, mantle, adductor muscle and labial palp) were collected from the same oyster used for genome sequencing. Total RNA was extracted separately using TRIzol reagent (OMEGA, USA) according to the manufacturer’s instructions. RNA quality and quantity were assessed using the Agilent 2100 instrument and Qubit 2.0 fluorometer (Thermo Fisher, USA), respectively. RNA from four tissues were combined in equimolar quantities into a single pool. Full-length coding DNA (cDNA) was obtained using SMARTer PCR cDNA Synthesis Kit (Clontech, USA). The library was constructed using SMRTbell® Template Prep Kit 2.0 (PacBio, USA). cDNA was sequenced by PacBio Sequel II platform with SMRT cell 8M (Biomarker, China).” (Lines 711-720)

We conducted k-mer copy number (KCN) analyses using $k = 17, 19$ and 21 to estimate genome size. K was the least odd number that conform to the following formula: $4^k/G >$

200, where G indicates the genome size (bp). Previous study estimated the genome size of *C. gigas* and *C. ariakensis* by flow cytometry, where their respective C-values, 0.89 pg and 0.99 pg⁶⁸, have a ratio of 1.112. Assuming the genome size of *C. gigas* is 586.8 Mb³⁴, the genome size of *C. ariakensis* would be 652.7 Mb. Thus, we selected 19-mer for genome size estimation. (Lines 646-652)

The genome size estimation by 19-mer analysis was 614.05 Mb, which was concordant with the final assembly size (613.89 Mb). We added it in the revised MS: “We produced a chromosome-level assembly of the estuarine oyster genome of an estimated size of 614.05 Mb (19-mer analysis) (Supplementary Fig. 1, Supplementary Table 1),” (Lines 92-94)

Supplementary Fig. 1 | Distribution of k-mer (k = 17, 19 and 21) frequency in the sequencing reads used to estimate genome size.

68 Xu, F. *Reproductive isolations between oysters of genus Crassostrea on the Xiaomiaohong oyster reef* Doctoral thesis, Institute of Oceanology, Chinese Academy of Sciences, (2009). (in Chinese)

The following was provided on Hi-C sequencing:

“The same adductor muscle was used for crosslink with formaldehyde at room temperature and DNA was digested with HindIII. The 5’ overhangs were adapted with biotinylated nucleotides, and free blunt ends were ligated. Then, the crosslinks were reversed and the DNA was purified and further sheared to 300-700 bp fragments. Streptavidin beads were used to isolate biotin-tagged fragments for PCR enrichment. The Hi-C library was constructed using Illumina’s paired-end kits according to the manufacturer’s instructions, which was subsequently sequenced on the Illumina HiSeq 4000 platform.” (Lines 669-676)

A total of 106.34 Gb Clean Data was generated for the HiC library, which was presented in the **Results** at line 96.

The 20 and 60 numbers mean the salinity. We corrected as: “20 ‰ and 60 ‰ for 7 days” (Line 823)

“we collected wild oysters and exposed them to different temperatures (20 °C and 37 °C for 6 hours) and salinities (20 ‰ and 60 ‰ for 7 days), respectively. The control oysters were exposed to seawater of 20 ‰ salinity at 20 °C.” (Lines 821-824)

Other very minor comments:

Fig. 1. Figure 1e is missing (duplicated gene cluster of the solute carrier families showing selection signals).

>>>Response: Sorry for the confusion. The ‘e’ indicated the innermost circle of **Fig. 1c**. We changed the location of ‘e’ to be clear in the revised MS. In addition, we provided the figure for duplicated gene cluster of the solute carrier families showing selection signals as Supplementary Fig. 13 and cited it in the revised MS.

Fig. 1c, CIRCOS plot showing 10 chromosomes (a), the distribution of GC content (b), transposable elements (c), coding sequences (d), and duplicated gene cluster of the *solute carrier* families showing selection signals (e, also see **Supplementary Fig. 13**).

Line 284: Only genes with 10 or more aligned reads in >90% samples were used for subsequent analyses -> what is the reason for this? Differential gene expression analyses are designed to take into account these factors if replicates are included in the study.

>>>Response: Although differential gene expression analyses take into account the factor for low expression genes, the method used in DEseq2 is to remove genes that the sum total expression of all samples was below a threshold, which otherwise would lead to false positives for genes only highly expressed in a few samples from different groups and false negatives for genes highly expressed in samples from a certain group. Thus, we removed these genes with low expression in more than 90% samples according to Kenkel & Matz (2016).

Kenkel, C. D. & Matz, M. V. Gene expression plasticity as a mechanism of coral adaptation to a variable environment. *Nature Ecology & Evolution* **1**, 0014, doi:10.1038/s41559-016-0014 (2016).

Line 289: delete 'and' in front of 75.9%

>>>Response: Corrected, thanks. (Line 309)

Line 400: I'd suggest rephrasing the sentence as follows: 'Expansion and selection in upstream regulatory regions of environment-responsive genes may hence be critical for the adaptation to rapidly changing environments in the estuarine oyster.' The part of enhancing the transcriptional complexity and phenotypic plasticity is not very thoroughly tested in this manuscript, although the results point to it.

>>>Response: We agree and revised it as suggested. (Lines 426-428)

Data availability: it is mentioned that the genome, whole genome resequencing and transcriptomic data are deposited in the cited BioProject. I strongly encourage the authors to deposit as well the genome annotations, since the gene repertoire is also

discussed in the manuscript.

>>>Response: We understand the point and uploaded the details for genome annotation of estuarine oyster, including gene locations (.gff3), protein models (.pep), CDS (.cds) and exon (.exon) sequences in FigShare online (https://figshare.com/articles/dataset/Genome_of_the_estuarine_oyster_provides_insights_into_climate_impact_and_adaptive_plasticity/16557390).

REVIEWERS' COMMENTS:

Reviewer #1 (Remarks to the Author):

The authors responded thoroughly to all suggestions and addressed all concerns. I appreciate the new figures and supplements, as well as the additional detail to the methods, and I'm sure that the gene models will facilitate others to use the data generated in future studies.

Reviewer #2 (Remarks to the Author):

The authors have addressed all my concerns. Congratulations on a very beautiful piece of work!